

# The first record of chigutisaurid amphibian from the Late Triassic Tiki Formation and the probable Carnian pluvial episode in central India

Sanjukta Chakravorti and Dhurjati Prasad Sengupta

Geological Studies Unit, Indian Statistical Institute, Kolkata, West Bengal, India

## ABSTRACT

A new, partially preserved skull of chigutisaurid amphibian (temnospondyli) has been reported for the first time from the Late Triassic Tiki Formation of India. Chigutisaurids are now known to occur in Australia's Early and Late Triassic, the Late Triassic in India, Argentina, and Brazil, the Jurassic of South Africa and Australia, and the Cretaceous of Australia. In India, the first appearance of chigutisaurids marks the Carnian—middle Carnian/Norian Boundary. This work also attempts to correlate, again for the first time, the advent of chigutisaurids and the occurrence of Carnian Pluvial Episodes (CPE) in the Late Triassic Maleri and Tiki Formation of Central India. The new specimen belongs to the genus *Compsocerops* prevalent in the Late Triassic Maleri Formation occurring 700 km south. However, the chigutisaurid specimen recovered from the Tiki Formation is a new species when compared to that of the Maleri Formation. It has the presence of an inward curved process of the quadratojugal as opposed to the straight downward trending process of the quadratojugal, the presence of vomerine foramen, shorter and wider interpterygoid vacuities, wider subtemporal vacuities, and the base of the interpterygoid vacuities at the same level with the base of the subtemporal vacuity. It proves that the Tiki Formation is coeval with the Lower Maleri Formation and a part of Upper Maleri.

## INTRODUCTION

Temnospondyls are a very diverse and widespread group of extinct amphibians that thrived from the Carboniferous to the Cretaceous reaching their peak in the Triassic (*Konietzko-Meier et al., 2018*; *Konietzko-Meier et al., 2019*). Capable of flourishing in both land and water in a variety of ecological niches, the fossils of these extinct amphibians are found in almost all places from Antarctica in the South to Greenland in the North, India being no exception (*Schoch, Milner & Witzmann, 2014*). The temnospondyl fossils in India are found in various Gondwana deposits throughout the country (*Chakravorti & Sengupta, 2019*; *Chowdhury, 1965*; *Sengupta, 1995*; *Sengupta, 2002*; *Tripathi, 1969*). Chigutisaurids are by far one of the most important temnospondyl amphibians, having the longest temporal range (*Marsicano, 1999*) ranging from the Early Triassic to the Cretaceous (*Dias-da Silva*

Corresponding author
Sanjukta Chakravorti,
chirpymoni2009@gmail.com

*et al., 2012*). However, the majority of them originated in the Upper Carnian to Lower Norian of the Late Triassic Period. The family Chigutisauridae is much less diversified than the other temnospondyl families. Falling within the superfamily Brachyopoidea which comprises parabolic, brevirostrine skulled temnospondyls, Chigutisauridae forms a single monophyletic family in the phylogenetic position (*Warren & Marsicano, 2000*). The spatial and temporal distribution of the chigutisauridae is schematically represented in Table 1 along with their general habitat (*Bandyopadhyay & Ray, 2020*; *Bonaparte, 1975*; *Cabrera, 1944*; *Dias-da Silva et al., 2012*; *Pledge, 2013*; *Rusconi, 1949*; *Rusconi, 1951*; *Sengupta, 1995*; *Warren, 1981*; *Warren & Marsicano, 2000*). The earliest origin of the family chigutisauridae is *Keratobrachyops australis* in the Early Triassic of Australia (*Warren, 1981*) with the latest being *Koolasuchus cleelandi* (*Warren, Rich & Vickers-Rich, 1997*) in the Cretaceous of Australia as well. So far, the origin and diversification of chigutisaurids remain restricted only to the Gondwana countries.

The Gondwana successions of India are exposed in four discrete basins coinciding with some of the major river valleys throughout the Indian subcontinent (*Pascoe, 1973*; *Robinson, 1970*; *Veevers & Tewari, 1995*). Of these, the Late Triassic Maleri and the Tiki Formations of the Pranhita-Godavari Valley Basin and the Son Valley Basin respectively are long known to be coeval (*Chatterjee, 1974*; *Chatterjee & Roy-Chowdhury, 1974*; *Mukherjee & Ray, 2014*; *Robinson, 1970*). Both Formations are known for the metoposaurid *Panthasaurus maleriensis* (*Arche & Lopez-Gomez, 2014*; *Chakravorti & Sengupta, 2019*; *Sengupta, 2002*). In the Late Triassic Maleri Formation *P. maleriensis* is thought to be restricted within the Carnian and the chigutisaurids appear in the mid-Carnian to early Norian (*Chakravorti & Sengupta, 2019*; *Sengupta, 1995*). Though a considerable amount of work has been done on the microvertebrates (*Bhat, 2017*; *Hussain, 2018*; *Ray et al., 2016*), rhynchosaurs (*Mukherjee & Ray, 2012*; *Mukherjee & Ray, 2014*) and phytosaurs (*Datta, Kumar & Ray, 2021a*; *Datta, Mukherjee & Ray, 2019a*; *Datta, Ray & Bandyopadhyay, 2021b*) of the Tiki Formation; no comprehensive work has been done in the last decade on its temnospondyl faunal contents. *Chakravorti & Sengupta (2019)* in their taxonomic revision of the Indian metoposaurids, included the metoposaurids of the Tiki Formation and grouped them into a new genus *Panthasaurus maleriensis* based on morphometric and phylogenetic approaches. However, the biostratigraphic implications of the Tiki Formation based on its temnospondyl contents have not been attempted to date. However, the taphonomic aspects of metoposaurids from the Tiki Formation have been discussed by *Rakshit & Ray (2020)*. Also, to date, no chigutisaurid remains were reported from the Late Triassic Tiki Formation though the same is widely prevalent in the Late Triassic Maleri Formation (*Sengupta, 1995*). Therefore, the finding of a chigutisaurid amphibian from the Late Triassic Tiki Formation is very important in the context of correlating the Late Triassic Maleri and Tiki Formation, India, and their position concerning global biostratigraphic correlation. This article will subsequently highlight a brief geological setting of the Tiki Formation followed by the taxonomic status of chigutisaurids from the Tiki Formation and subsequently its role in demarcating the Carnian Pluvial Episode in India.

**Table 1** The table shows the spatial and temporal range of the family Chigutisauridae along with their paleoecology and paleoenvironment.

| Name | Early_interval | Late_interval | Max_ma | Min_ma | Country | Fm. | Lithology | Palaeoenvironment |
|---|---|---|---|---|---|---|---|---|
| *Koolasuchus cleelandi* | Late Aptian | _ | 122.46 | 112.03 | Australia | Eumeralla Fm. | Sandstone | |
| *Koolasuchus cleelandi* | Late Aptian | _ | 122.46 | 112.03 | Australia | Eumeralla Fm. | Sandstone | |
| *Koolasuchus cleelandi* | Late Aptian | _ | 122.46 | 112.03 | Australia | Eumeralla Fm. | Sandstone | |
| *Koolasuchus cleelandi* | Late Aptian | _ | 122.46 | 112.03 | Australia | Eumeralla Fm. | Sandstone | |
| *Koolasuchus cleelandi* | Aptian | _ | 125 | 113 | Australia | Eumeralla Fm. | Sandstone | |
| *Siderops kehli* | Pliensbachian | Toarcian | 190.8 | 174.1 | Australia | Evergreen Fm. | Ironstone/ Sandstone | |
| *Chigutisauridae indet* | Mid Carnian | _ | 228 | _ | Australia | Leigh Creek Fm. | Siltstone | |
| *Compsocerops* | Mid Carnian | Norian | 228 | 208.5 | Brazil | Santa Maria Fm. | Mudstone | |
| *Compsocerops cosgriffi* | Mid Carnian | Norian | 228 | 208.5 | India | Upper Maleri Fm. | Mudstone | |
| *Kuttycephalus triangularis* | Mid Carnian | Norian | 228 | 208.5 | India | Upper Maleri Fm. | Mudstone | |
| *Compsocerops tikiensis* | Mid Carnian | Norian | 228 | 208.5 | India | Upper Tiki Fm | Mudstone | |
| *Pelorocephalus mendozensis* | Carnian | Norian | 237 | 208.5 | Argentina | Cacheuta Fm. | "Siliciclastic" | The overall palaeoenvironment |
| *Pelorocephalus tenax* | Carnian | Norian | 237 | 208.5 | Argentina | Cacheuta Fm. | "Siliciclastic" | is fluviolacustrine |
| *Pelorocephalus cacheutensis* | Carnian | Norian | 237 | 208.5 | Argentina | Cacheuta Fm. | "Siliciclastic" | |
| *Pelorocephalus ischigualastensis* | Carnian | Norian | 237 | 208.5 | Argentina | Ischigualasto Fm. | Siliciclastic | |
| *Keratobrachyops australis* | Induan | _ | 252.17 | 251.2 | Australia | Arcadia Fm. | Mudstone | |

**Notes.**

Max ma, Maximum age million years ago; Min ma, Minimum age million years ago; Fm., Geological Formations from which the specimen had been excavated.

The table has been modified after data was acquired from the Paleobiology Database on 10 th May 2022, using the family name "Chigutisauridae". Data for the table was modified after *Bonaparte (1975)*, *Marsicano (1999)*, *Dias-da Silva et al. (2012)*, *Warren (2006)*, *Pledge (2013)*, *Cabrera (1944)*, *Rusconi (1949)*, *Rusconi (1951)*, *Sengupta (1995)*, *Warren & Hutchinson (1983)*, *Warren (1981)* and *Warren & Marsicano (2000)*.

## The Carnian Pluvial Episode—a global climatic consequence

The Carnian Pluvial Episode (CPE) can be defined as a geologically short-lived (234–232 million years ago) monsoonal period of extreme rainfall that brought about significant changes in several depositional environments (*Arche & Lopez-Gomez, 2014*; *Dal Corso et al., 2015*; *Furin et al., 2006*; *Schlager & Schöllnberger, 1974*; *Simms & Ruffell, 1990*). The Carnian Pluvial Episode (CPE) was a global phenomenon. Geochemical data suggest that global warming involved environmental and biotic changes. Radioisotopic ages coupled

with biostratigraphic correlation suggest a possible link to the eruption of the Wrangellia Large Igneous Province (LIP) (*Dal Corso et al., 2020*). CPE was a significant (but previously neglected) time of extinction linked to the Carnian explosive diversification of many key modern groups of plants and animals (*Dal Corso et al., 2020*). The CPE marks a distinct change in the hydrological cycle during which the climate shifted from arid to humid conditions and back again to arid conditions (*Bernardi et al., 2018*). It is represented by remarkable enhancement of the hydrological cycle demarcated by four episodes of increased rainfall indicated by diverse sedimentary and paleontological data, repeated Carbon Cycle perturbations, evidenced by sharp negative C-isotope excursions, coincided with global environmental changes and climate warming all of which suggest a cause-and-effect relationship (*Dal Corso et al., 2015*).

The Carnian is the earliest part of the Late Triassic and its base or lower boundary is dated at approximately 237 million years based on U-Pb radiometric dating of a single crystal zircon from a tuff layer within a section having strong biostratigraphic constraints (*Dal Corso et al., 2015*; *Dal Corso et al., 2012*; *Maron et al., 2019*). The upper boundary of the Carnian is constrained at approximately 227 million years based on magnetostratigraphic correlations between the marine successions of Tethys and the astrochronological time scale of the continental Newark Basin (*Kent, Olsen & Muttoni, 2017*). The Carnian is subdivided into Julian (Early Carnian) and Tuvalian (Late Carnian) substages. The Julian–Tuvalian boundary occurs at approximately 233 million years (*Dal Corso et al., 2015*; *Kent, Olsen & Muttoni, 2017*). The beginning of the onset of CPE is well defined from ammonoid, conodont, and sporomorph biostratigraphic dating and is synchronous in several geological settings. It coincides with the first appearance of the ammonoid genus *Austrotrachyceras* in the Julian (*Dal Corso et al., 2020*; *Dal Corso et al., 2012*; *Roghi et al., 2010*; *Simms & Ruffell, 1990*; *Sun et al., 2016*). However, the upper boundary or the end of CPE is poorly defined in most locations. It is usually placed at the base or within the Tuvalian 2 based on sedimentological (*e.g.*, end of terrigenous sediment supply) and chemostratigraphic (last C-isotope excursion) evidence (*Dal Corso et al., 2020*; *Dal Corso et al., 2015*; *Dal Corso, Ruffell & Preto, 2018*). The total duration of this pluvial episode is variable. Cyclostratigraphy of marine successions of the South China Block and continental successions of the Wessex Basin (United Kingdom) gives a duration of the CPE of approximately 1.2 ma but this is variable and longer than 1.6−1.7 million years as indicated by integrated stratigraphy (biostratigraphy and magnetostratigraphy).

## Temnospondyl amphibians in the carnian pluvial episode

The CPE facilitated the Dinosaur Diversification Event (DDE) (*Bernardi et al., 2018*). However, the role of CPE on the temnospondyls has not much been discussed barring a few articles (*Buffa, Jalil & Steyer, 2019*; *Fortuny et al., 2019*; *Gee & Jasinski, 2021a*; *Lucas, 2020*). The amphibious temnospondyls living both on land and water were the most sensitive to the changes in climate. Two dominant groups of temnospondyls, in this context, were the metoposaurids and the chigutisaurids. According to *Fortuny et al. (2019)* the gigantism of the metoposaurids might have been linked to the Carnian Pluvial Episode. *Buffa, Jalil & Steyer (2019)* also stated that the diversification of the metoposaurids might

have been linked to the CPE and the post-CPE aridification led to the extinction of the metoposaurids during the Rhaetian. *Gee & Jasinski (2021a)* have also commented on the fact that the physiological variation of the metoposauridae and their palaeoclimatic range also corroborates a palaeo-environmental barrier. Finally, *Lucas (2020)* concluded that climate change that occurred during CPE played an important part in the metoposaurid evolution. According to *Lucas (2020)*, Metoposaurids appeared during the CPE, attained their highest diversity and cosmopolitan distribution during this time and had reduced diversity and showed endemism in the post-CPE climate.

Thus, the presence of *Compsocerops* in both Maleri and Tiki Formation enhances the scope to discuss the palaeoenvironment of these two Late Triassic basins in India and to compare the possible reason for faunal turnover from Carnian to Norian concerning the amphibious temnospondyls (*Sengupta, 1995*).

## GEOLOGICAL SETTING OF THE TIKI FORMATION AND ITS COMPARISON TO THE MALERI FORMATION

The Tiki Formation named after the small village of Tiki in the district of Shahdol, Madhya Pradesh has been an interest to scientists for decades. Reports on the geology and palaeontology of the Tiki Formation date back to as early as 1877 when *Hughes (1877)* noticed reptilian fossils near this village. *Cotter (1917)* noticed several other such fossils and finally *Fox (1931)* formally designated the area as the "Tiki stage". *Aiyengar (1937)* first divided the "Tiki stage" into lithostratigraphic units viz. the lower unit being fossiliferous and composed dominantly of red and green mudstones, proportionately lesser number of sandstones and mud-galls while the upper unit is composed of ferruginous sandstones and shales. *Robinson (1970)* in her memoir kept the Tiki Formation to be coeval with the Late Triassic Maleri Formation. However, *Dutta & Ghosh (1993)* did not recognize the separate entity of the Tiki Formation and placed Tiki rocks in the upper part of the Pali Formation forming the Pali-Tiki Formation. *Roychowdhury et al. (1975)*, based on the megaflora assemblage, noted the age of the Nidpur beds is Anisian and separated the upper part of the Tiki Formation to be Carnian –Rhaetian in age. *Maheshwari, Kumaran & Bose (1976)* separated the Tiki Formation to be a separate entity (including the Nidpur beds) and based on the mega flora and faunal assemblages suggested the age of the Tiki (including Nidpur beds) Formation to be ranging from Anisian to Norian with a possible extension to Rhaetian. *Mukherjee et al., (2012)* revised the stratigraphy of the Rewa Basin and put the Tiki Formation with the coeval Carnian Lower Maleri Formation. *Ray et al. (2016)* in the study of vertebrate faunal assemblage of the Tiki formation also suggested Tiki Formation be of Carnian in age but they narrowed the range to Otischalkian to early Adamanian. The common conclusions of all this literature are that the Late Triassic Tiki Formation is dominantly Carnian and its fauna can be correlated with the Lower Maleri fauna. So far, no evidence of a Norian age was assigned to any part of the Tiki Formation. As stated earlier, based on the faunal pieces of evidence and correlating it with the Late Triassic Maleri Formation of the Pranhita–Godavari valley a Carnian age was assigned to the Tiki Formation (*Dutta & Ghosh, 1993*; *Kutty, Jain & Chowdhury, 1987*; *Sengupta,*

*1992*). Henceforth, through the decades, the Tiki formation was considered to be coeval with the Carnian Maleri Formation (*Mukhopadhyay et al., 2010*; *Sengupta, 1992*; *Veevers & Tewari, 1995*). Only recently, *Datta, Ray & Bandyopadhyay (2021b)* while describing a new phytosaur from the Tiki Formation, commented that the age of the Tiki Formation may range from Carnian to Early/Middle Norian.

To date, the faunal assemblage of the Tiki Formation includes fishes belonging to the family Ceratodontidae, Hybodontidae and new undescribed forms of Xenacanthidae (*Ray et al., 2016*), temnospondyl amphibians belonging to Metoposauridae; reptilian belonging to families Rhyncosauridae, Rauisuchidae, Acrodanta, basal Saurischia, Dromatheridae, and Traversodontidae. Mammaliaformes are also reported from the Tiki Formation (Tables 2A and 2B) (*Bandyopadhyay & Ray, 2020*; *Ray et al., 2016*).

The appearance of chigutisaurids in India is noted with the demise of the metoposaurs, rhynchosaurs, and primitive phytosaurs. Prosauropods, large in size also appeared during that time (*Novas et al., 2010*). These faunal turnovers were thought to demarcate India's Carnian–Norian boundary (*Datta, Ray & Bandyopadhyay, 2021b*). However, recent signatures (*Dal Corso et al., 2015*) of the pluvial event and its role in extinction might shift this boundary to Carnian–mid-Carnian/Norian.

The Maleri Formation starts with a 250-meter-thick mudstone (*Dasgupta, Ghosh & Gierlowski-Kordesch, 2017*; *Kutty & Sengupta, 1989*). At the top of the mudstone, a sandy zone initiates the sand–mud alternations of Upper Maleri (*Kutty & Sengupta, 1989*). This sandy zone contains a maximum number of rhynchosaur fossils, abundant metoposaurids, and unionids. The chigutisaurids in Maleri appear just above this sandy zone (*Sengupta, 1995*) and no rhynchosaurs or metoposaurids are known from that level (or above that). The occurrence of chigutisaurids in upper part of Tiki Formation is also restricted within a sandy zone which do not contain metoposaurids or rhynchosaurs. Unionids are also present there but in lesser abundance than Maleri. Phytosaur teeth are also present. This sandy horizon noticed in Maleri and Tiki has been stratigraphically placed below the Carnian–mid-Carnian/ Norian boundary and may indicate the Carnian Pluvial Episodes (CPE) in India.

## MATERIALS USED WITH AN OVERVIEW OF THEIR PRESERVATION AND METHODS

The new skull material along with the clavicle (ISI A 202) excavated by the authors and the specimen RH01/Pal/CHQ/Tiki/15 previously described as metoposaurid (*Kumar & Sharma, 2019*) has been studied in detail and referred to in this article. The map of the temnospondyl-bearing localities of the Tiki Formation has been modified here with faunal boundaries (hypothetical faunal boundary demarcated in a red dotted line, after *Mukherjee & Ray, 2012*) (Fig. 1). The temnospondyl bearing (metoposaurid and chigutisaurid) localities of the Maleri Formation have also been extensively mapped and modified (after *Dasgupta, Ghosh & Gierlowski-Kordesch, 2017*; *Kutty & Sengupta, 1989*) (Fig. 2). Some distinct sections have been logged in the Tiki Formation and has been compared with the existing and modified logs of the Late Triassic Tiki and Maleri Formation.

**Table 2** Fossils excavated from the Late Triassic Tiki Formation (modified from *Bandyopadhyay & Ray, 2020*).

**2a. Fish fossils excavated from the Late Triassic Tiki Formation (modified after *Bandyopadhyay & Ray, 2020*).**

| Order/Family Chondrichthyes | Genus and Species | Order/Family Osteichthyes | Genus and Species |
|---|---|---|---|
| Lonchididae | *Lonchidion estesi* | Ptychoceratodontidae | *Ceratodus sp.* |
|  | *Lonchiodon incumbens* |  | *Ptychoceratodus oldhami* |
|  | *Pristrisodus tikiensis* | Gnathorhizidae | *Gnathorhiza sp.* |
| Xenacanthidae | *Mooreodontus indicus* | Actinopterygii | Undescribed |
|  | *Mooreodontus jaini* |  |  |
|  | *Tikiodontus asymmetricus* |  |  |

**2b. Vertebrate fossil assemblage (tetrapod content) of the Late Triassic Tiki Formation of the Rewa Basin, India (modified after *Bandyopadhyay & Ray, 2020*).**

| Order/Family Amphibia | Genus and Species | Order/Family Diapsida | Genus and Species |
|---|---|---|---|
| Metoposauridae | *Panthasaurus maleriensis* | Archosauriformes | *Galtonia sp.*, *Protecovasaurus sp.*, and other intermediate forms |
| Chigutisauridae | *Compsoceraps tikiensis* | Dinosauriformes | UndescribedTheropod-like (?) forms |
| **Diapsida** |  | Aetosauria | Undescribed |
| Phytosauria | *Volcanosuchus statisticae(?) leptosuchomorph* | Synapsida |  |
| Rhynchosauria | *Hyperodapedon tikiensis* | Cynodontia | *Ruberodon roychowdhurii* |
| Rauisuchidae | *Tikisuchus romeri* | Mammaliaformes | *Tikitherium copei* |
| Rhynchocephalia | *Undescribed* |  | *Gondwanadon tapani* |

*Preservation of specimen ISI A 202.* The skull along with a fragmentary clavicle, ISI A 202, is poorly preserved (Figs. 3, 4, 5 and 6). Only the left half of the skull is preserved and the specimen is heavily eroded. Thus, the ornaments are not well observed in all the areas. The upper part of the parietal and postfrontal have coarse ridges and grooves preserved in them. The skull, its fragments and the clavicle, all have been excavated from the same point in the location and were present together with the same individual as the skull.

*Preservation of specimen RH01/Pal/CHQ/Tiki/15.* Only the picture of the palate published as *Metoposaurus* in the article published by *Kumar & Sharma (2019)* (Fig. 7) could be studied. As mentioned in the article (*Kumar & Sharma, 2019*) the material could not be excavated from the field. The photograph of the said publication was reproduced with permission of the journal editor and a higher resolution image was reproduced for better clarity and study. The palate is dorsoventrally elongated and slightly sheared. The edges of the palate are not well preserved.

The new specimens with ISI numbers, ISI A 202 and the published specimen of *Kumar & Sharma (2019)* RH01/Pal/CHQ/Tiki/15 were recovered from mudrocks at a distance of about 100 m from each other from the village of Tenduadh in the Late Triassic Tiki Formation. Thus, ISI A 202 and RH01/Pal/CHQ/Tiki/15 (previously published by *Kumar & Sharma, 2019*) as metoposaurid are now the two chigutisaurid individuals that are being reported for the first time from the Tiki Formation.

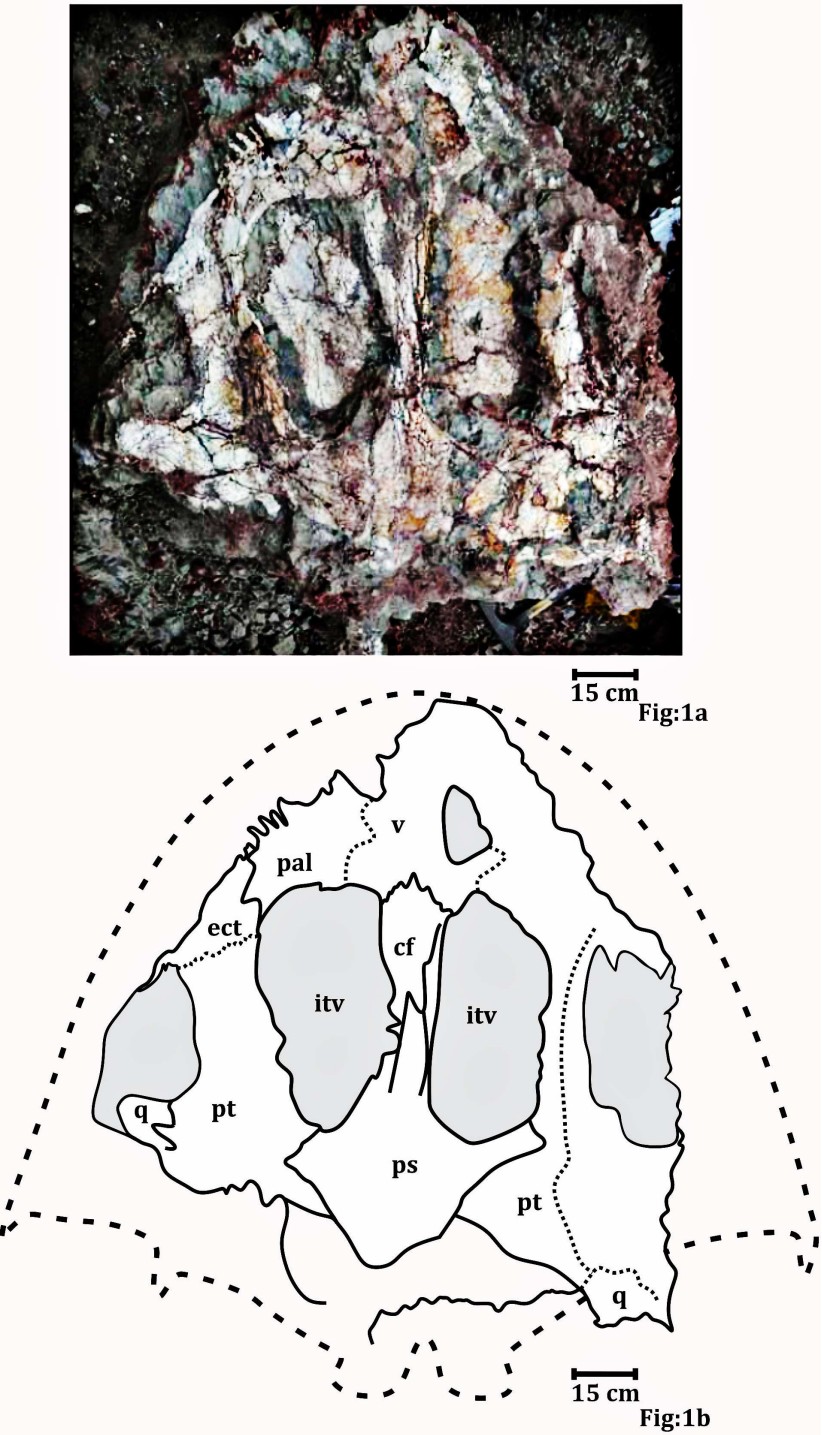

**15 cm**
**Fig:1a**

**15 cm**
**Fig:1b**

**Figure 1  Palatal surface of the skull photograph (RH01/Pal/CHQ/Tiki/15)** (*Kumar & Sharma, 2019*).
(A) The field photograph published in *Kumar & Sharma (2019)*. (B) Line drawing showing the disposition of the bones in the palatal surface of the skull published in *Kumar & Sharma (2019)*. The abbreviation stated in the figure are as follows: cf, cultriform process; ect, ectopterygoid; itv, interpterygoid vacuity; pal, palatine; ps, parasphenoid; q, quadrate; stf, subtemporal foramen; v, vomer. Recreated with permission from the editor of the Palaeontological Society of India.

The electronic version of this article in Portable Document Format (PDF) will represent a published work according to the International Commission on Zoological Nomenclature (ICZN), and hence the new names contained in the electronic version are effectively published under that Code from the electronic edition alone. This published work and the nomenclatural acts it contains have been registered in ZooBank, the online registration system for the ICZN. The ZooBank LSIDs (Life Science Identifiers) can be resolved and the associated information viewed through any standard web browser by appending the LSID to the prefix http://zoobank.org/. The LSID for this publication is urn:lsid:zoobank.org:pub:1B45D1E1-9FFE-421D-8060-4174334A7EF4. The online version of this work is archived and available from the following digital repositories: PeerJ, PubMed Central SCIE, and CLOCKSS.

# CHIGUTISAURIDAE FROM THE TIKI FORMATION
## Systematic Palaeontology

Temnospondyli *Von Zittel (1887)*
Stereospondyli *Von Zittel (1888)*
Chigutisauridae *Rusconi (1951)*
*Compsocerops Sengupta (1995)*

*Compsocerops tikiensis* sp. *nov.* (ISI A 202/1, ISI A 202/2, ISI A 202/3, ISI A 202/4, and ISI A 202/5, RH01/Pal/CHQ/Tiki/15) (Fig. 3).

**Type material**: ISI A 202/1 which comprises the left half of a skull roof, ISI A 202/2—a nearly complete clavicle and ISI A 202/3, ISI A 202/4, and ISI A 202/5—broken parts of the skull are the holotype. The holotype materials are housed in Geological Studies Unit, Indian Statistical Institute, Kolkata, India (Fig. 3).

**Paratypes:** ISI A 202/2, ISI A 202/3, ISI A 202/4 and ISI A 202/5, RH01/Pal/CHQ/Tiki/15.

**Referred material**: A palate (RH01/Pal/CHQ/Tiki/15) previously assigned to a metoposaurid by *Kumar & Sharma (2019)* has been referred (Fig. 7). The parabolic skull outline, vaulted pterygoid, shape and proportion of the interpterygoid vacuities, wide and folded palatine ramus of the pterygoid, and comparatively narrow cultriform process of the parasphenoid in RH01/Pal/CHQ/Tiki/15 indicate that it is not a metoposaurid. The cultriform process is wider than *Compsocerops cosgriffi* but narrower than any of the metoposaurids. This palate is comparatively well preserved and bears definite characteristics of a chigutisaur as it appears from the field photograph (Fig. 7). Only the photograph of the palate is available for study.

**Locality:** Southwest of the Village of Tenduadh (23°59′41″N; 81°25′02″E), just next to the Barakachh–Beohari Road in the district of Shahdol, Madhya Pradesh, Central India.

**Etymology**: The new species of chigutisaur is named after the Late Triassic Tiki Formation from where the specimen has been excavated and studied.

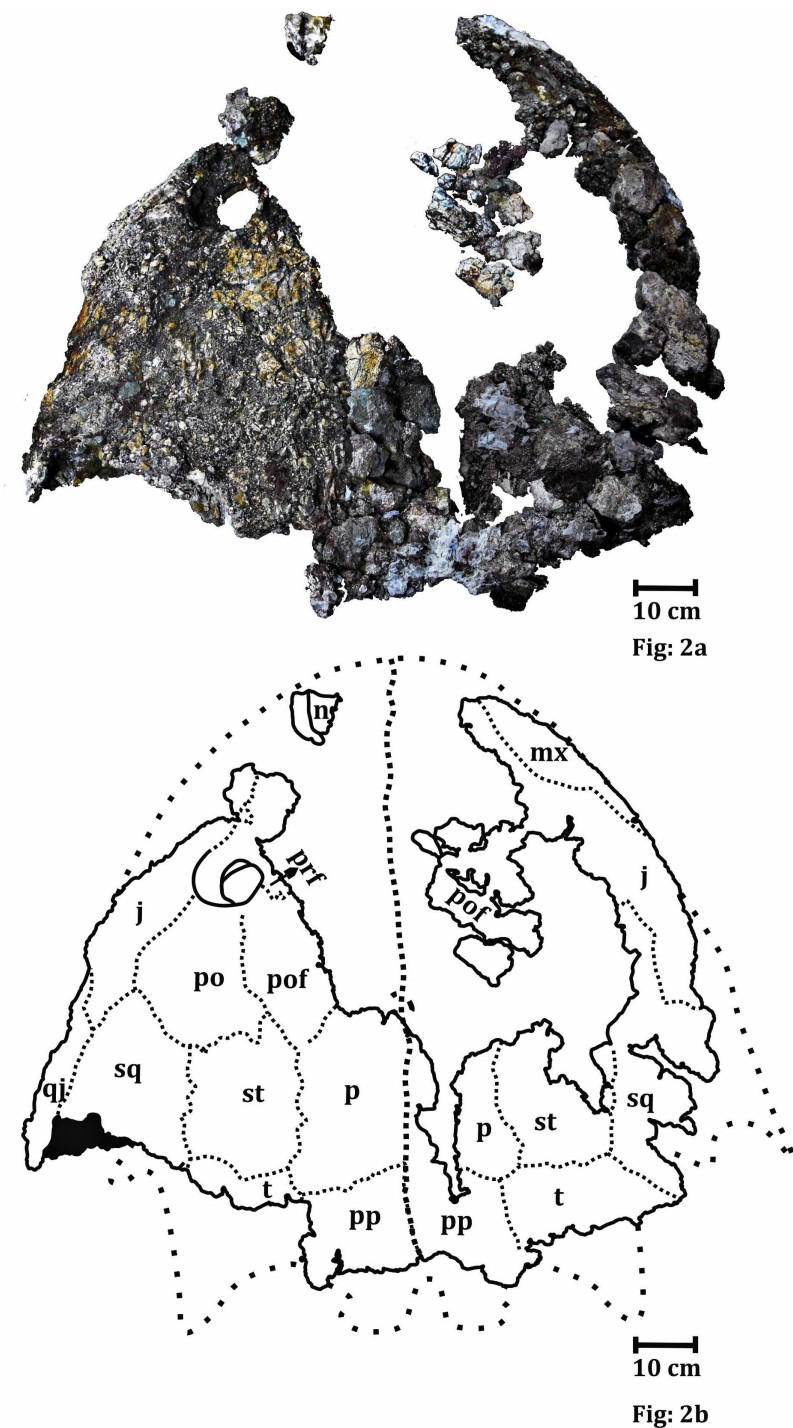

**Figure 2** **Dorsal surface of the skull roof of ISI A 202 *Compsocerops tikiensis* sp. nov.** (A) Reconstructed photograph of the dorsal surface of the skull roof of ISI A 202. Scale bar = 5 cm. (B) Line drawing showing the disposition of the preserved bones in the dorsal part of the skull roof in ISI A 202. The abbreviation stated in the figure are as follows: j, jugal; mx, maxilla; n, nasal; p, parietal; po, postorbital; pof, postfrontal; pp, postparietal; prf, prefrontal; qj, quadratojugal; sq, squamosal; st, supratemporal; t, tabular. Scale bar = 5 cm.

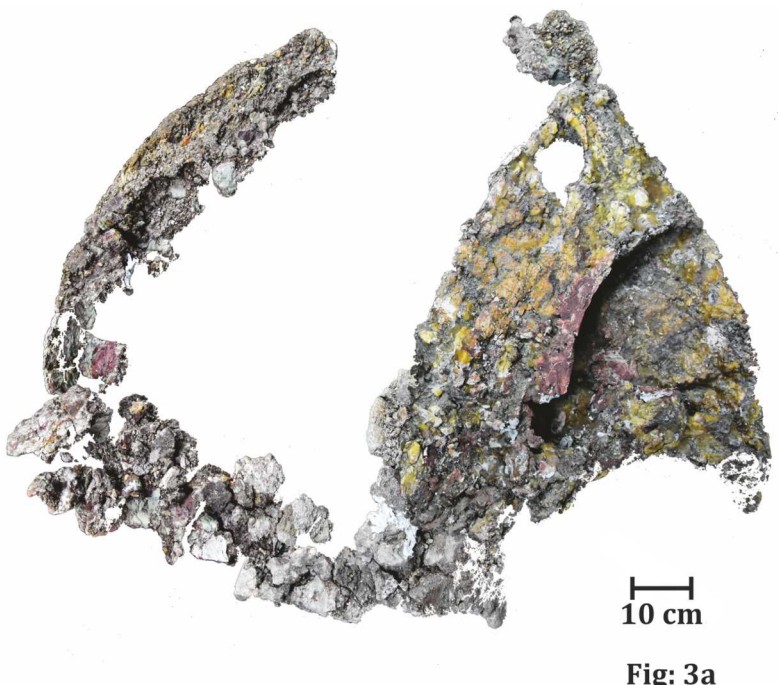

**Fig: 3a**

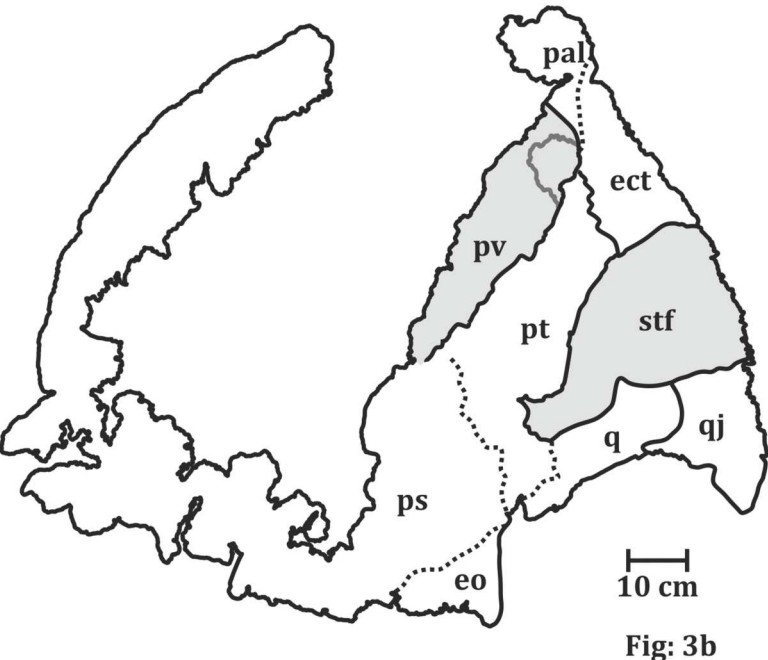

**Fig: 3b**

**Figure 3** **Palatal surface of the skull of ISI A 202 *Compsocerops tikiensis* sp. nov.** (A) Reconstructed photograph of the palatal surface of the skull of ISI A 202. Scale bar = 10 cm. (B) Line drawing showing the disposition of the preserved bones in the dorsal part of the skull roof in ISI A 202. The abbreviation stated in the figure are as follows: ect, ectopterygoid; eo, eoccipital; pal, palatine; ps, parasphenoid; pt, pterygoid; pv, palatine vacuity; q, quadrate; stf, subtemporal foramen.

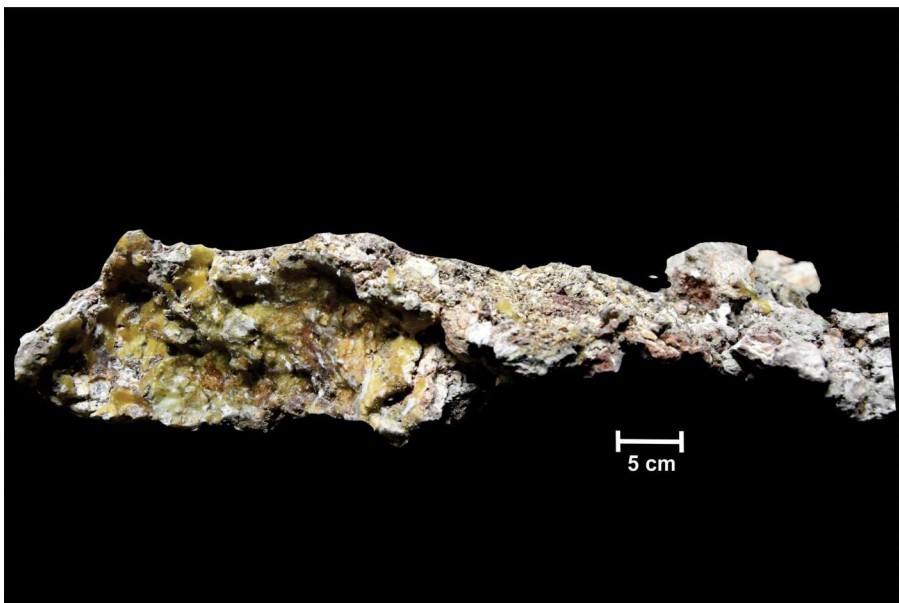

**Figure 4 The poorly preserved occiput of ISI A 202 *Compsocerops tikiensis* sp. nov.**

**Diagnosis of genus**: A chigutisaur temnospondyl identified as *Compsocerops* by the presence of the following combination of characters: skull outline parabolic in shape, orbits anteriorly placed, raised rim of the naris, presence of a well preserved conspicuous quadratojugal projection, presence of squamosal horn (which though broken and eroded is recognizable), and ill preserved postparietal horn (though ill preserved the presence of horns can be clearly identified), well preserved vaulted pterygoid, long and narrow cultriform process of the parasphenoid, dentigerous area restricted to the anterior portion of the palate, short and restricted palatine dentition not reaching to the middle of the choana, wide ramus of the pterygoid with a gentle fold, walls like quadrate ramus of the pterygoid, presence of postpterygoid process, typical shape of the ramus of the pterygoid and that of the subtemporal vacuities, long dorsal process of the clavicle with a distinct groove and bean-shaped cross section.

**Diagnosis of species**: The new species of *Compsocerops* is identified by the presence of an inward curved process of the quadratojugal as opposed to the straight downward trending process of the quadratojugal in *Compsocerops cosgriffi*, presence of vomerine foramen, shorter and wider interpterygoid vacuities, wider subtemporal vacuities, the base of the interpterygoid vacuities at the same level with the base of the subtemporal vacuity as opposed to *Compsocerops cosgriffi* (where the base of the interpterygoid vacuity is lower than the base of the subtemporal vacuity making the interpterygoid vacuities longer and slenderer in *Compsocerops cosgriffi*) and wider cultriform process of the parasphenoid.

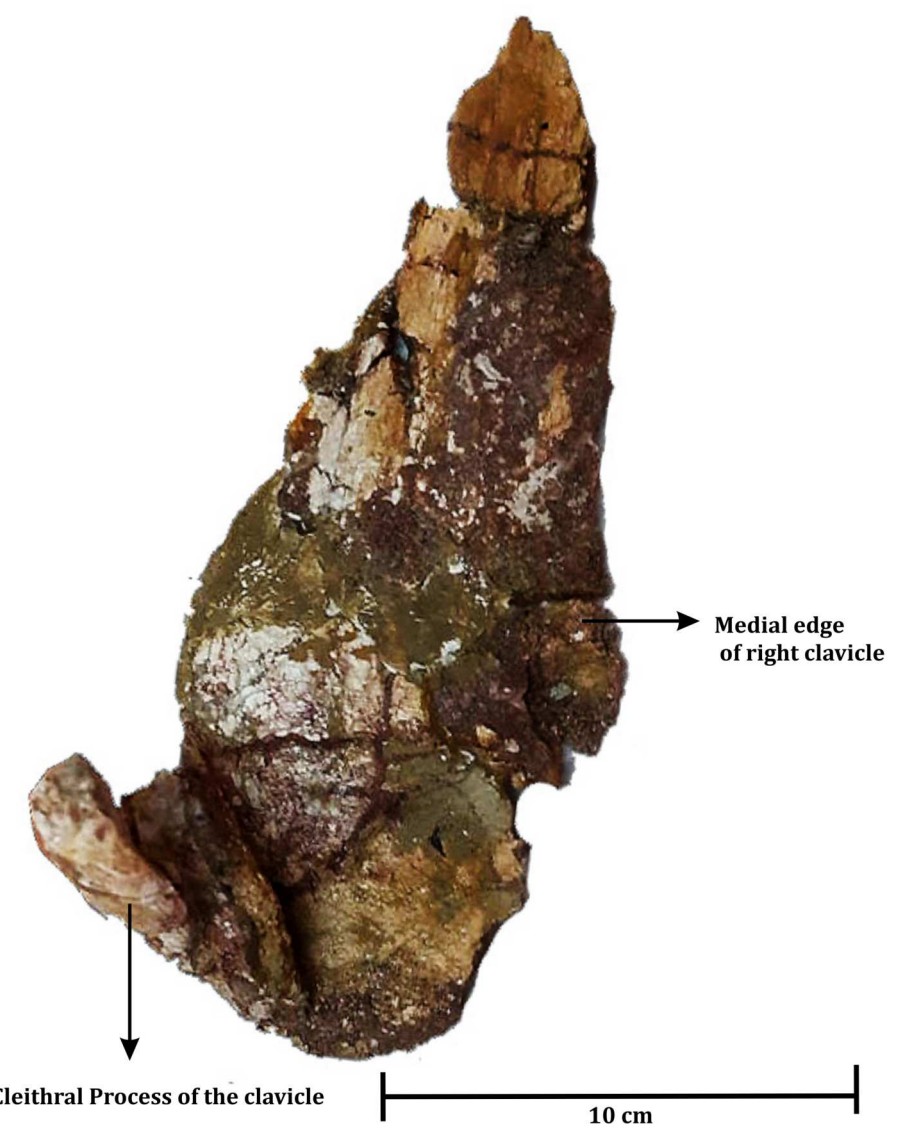

**Medial edge
of right clavicle**

**Cleithral Process of the clavicle**

10 cm

**Figure 5** Dorsal view of poorly preserved left clavicle of ISI A 202 *Compsocerops tikiensis* sp. nov.

## DESCRIPTION AND COMPARATIVE ANATOMY

**The skull roof** (Fig. 3)

The dorsal part of the skull roof can only be studied in ISI A 202.

The skull roof is parabolic in shape with a broad and concave posterior part of the skull table. Even though the anterior part of the skull roof is mostly broken, a major portion of the left orbit is preserved. The orbit is subcircular in shape and bordered by the prefrontal, jugal, postfrontal and postorbital. The orbit is located in the anterior half of the skull and is laterally placed. The posterior and posterolateral part of the left external nares is also preserved. It can be understood from the posterior outline of the external nares that they are oval. The external nares are located very close to the anterior border of

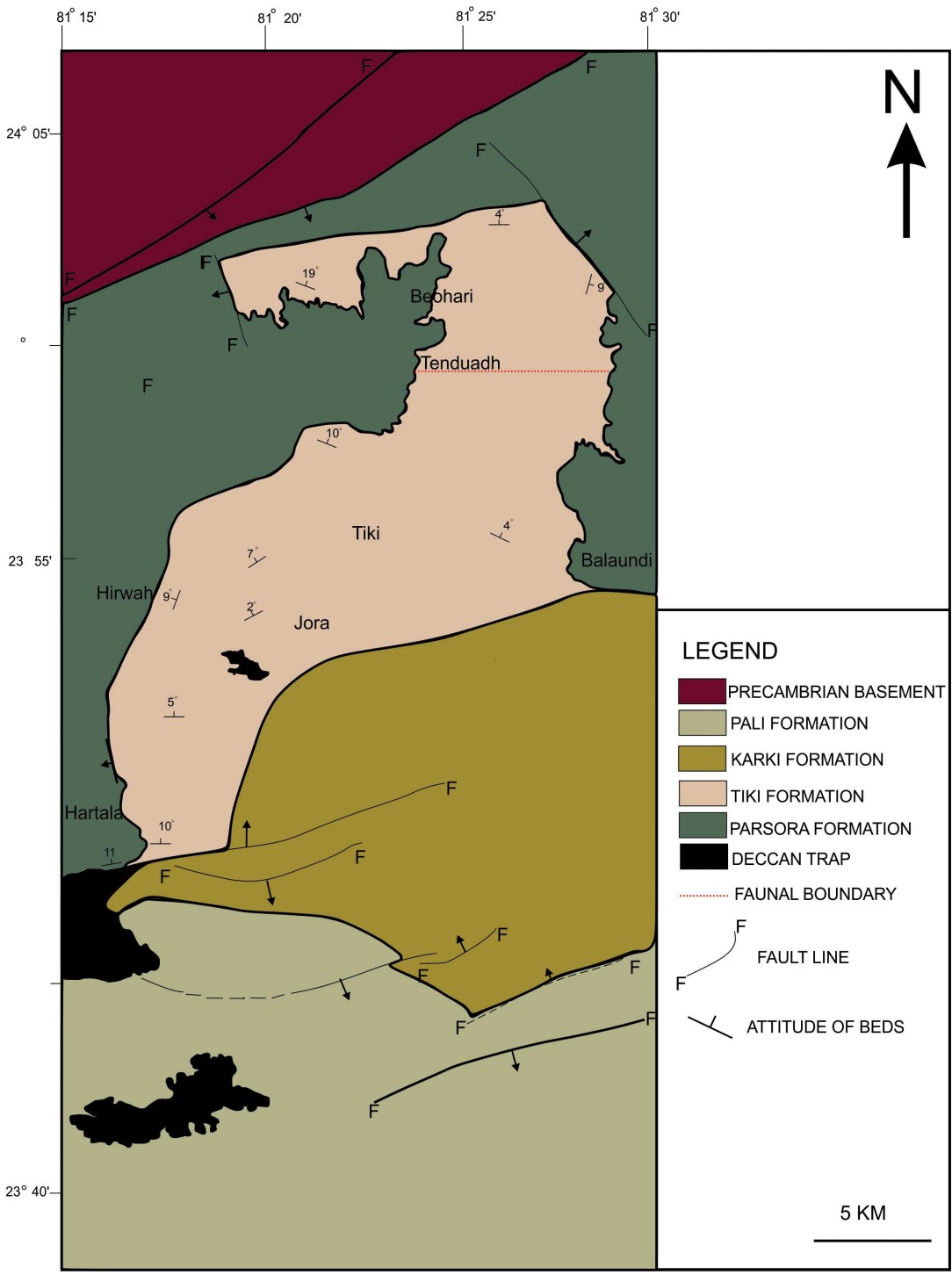

**Figure 6 Geological map showing the Tiki Formation, Rewa Basin, India.** Geological map of the Tiki Formation, Rewa Basin, India, modified from *Mukherjee et al. (2012)*. The red dotted line shows the hypothetical faunal boundary.

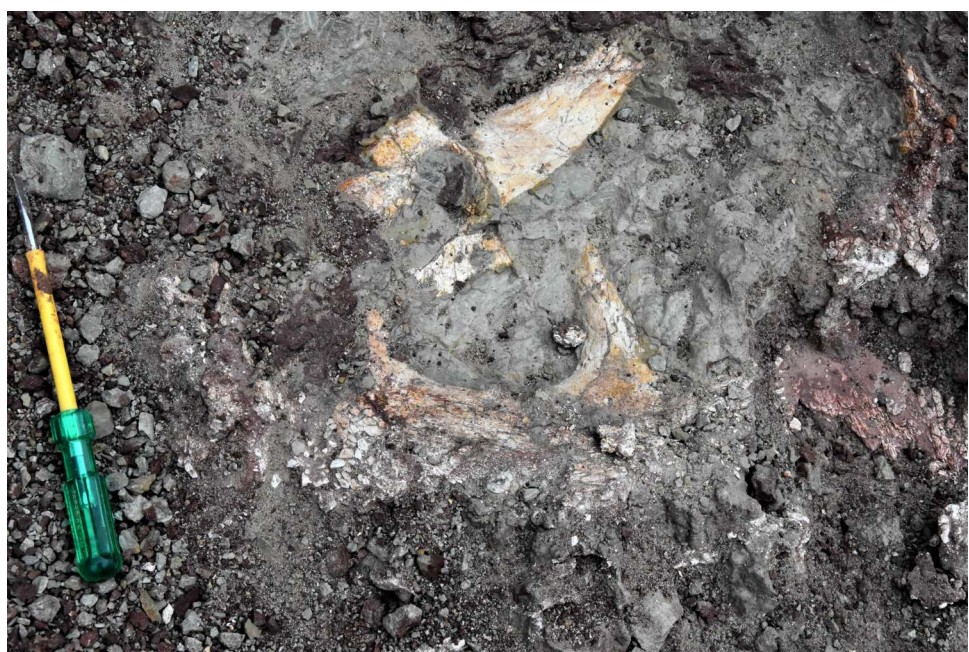

**Figure 7** **Field photograph showing the exposure of *Compsocerops tikiensis* sp. nov. embedded in mudstone in the Upper part of the Tiki Formation.** Image taken by the authors during field season 2019.

the skull roof. The posterior part of the external nares is thick and raised. This character is noted in chigutisaurids and is found in *Pelorocephalus tenax* (*Marsicano, 1999*) and *Compsocerops cosgriffi* (*Sengupta, 1995*). The supraorbital sensory canal in the region of the naris is unusually deep around the posterolateral border of the naris which is responsible for the thick and raised posterior part of the naris in the ISI A 202. This characteristic feature has also been noted in *Compsocerops cosgriffi* (*Sengupta, 1995*) where anteriorly the naris is flushed with the skull roof. The entire disposition of the sensory sulci is not well preserved in the specimen. Apart from the supraorbital sulcus, the presence of postorbital dermal sensory sulcus through the postfrontal can be recognized by the deep continuous canal like structure in these two bones. The infraorbital sulcus is visible in the maxilla but gradually becomes less prominent as it enters the jugal to form the jugal lateral dermal sensory sulcus. Just like other comparable chigutisaurids like *Keratobrachyops australis* (*Warren, 1981*), *Siderops kehli* (*Warren & Hutchinson, 1983*), *Pelorocephalus tenax* (*Marsicano, 1999*; *Rusconi, 1949*), *Compsocerops cosgriffi* (*Sengupta, 1995*) the lacrimal is absent in ISI A 202 and the maxilla enters the border of the external nares. However, the anterior part of the skull is fragmentary and heavily eroded. The better-preserved left side of the dorsal part of the skull roof consists of partially preserved prefrontal, postfrontal, postorbital and supratemporal. The squamosal is broken at the posterior part, and parietal, postparietal, tabular, jugal and quadratojugals are also partially preserved. The surfaces of the bones are eroded in most places and ornamentations are poorly preserved. The parietal is comparatively large, rectangular and broken along the midline. Coarse ridges and grooves can be recognized from the anterior part of the parietal. Just as in *Compsocerops*

the postparietal of ISI A 202 (Fig. 3) is much shorter in length than the parietal. The pineal foramen is not preserved in the parietal. The suture of the postparietal with the tabular is obliterated. The postparietal is broken and eroded along the midline and at its posterior part in the region of the postparietal horn. The postparietal horn is broken in ISI A 202 (Fig. 3). However, there is clear evidence that the horns exist. Postparietal horns are the most unambiguous synapomorphy of *Compsocerops*. It is preserved in *C. cosgriffi* (*Sengupta, 1995*), *C. sp.* (*Dias-da Silva, Cabreira & Da Silva, 2011*) and *C. tikiensis* sp. nov. These horns are not preserved in any other chigutisaur (the relevant area is not preserved in *Siderops Warren & Hutchinson, 1983* and *Koolasuchus Warren, Rich & Vickers-Rich, 1997*). The tabular is most likely to be in contact with the parietal though that part is not very well preserved. The broad tabular–parietal contact is considered to be a diagnostic character of *Compsocerops* (*Sengupta, 1995*). The tabular horn is broken. This post-quadratojugal process is robust and despite the very poor preservation of the skull in general, the posterior quadratojugal process is well preserved. The shapes and sutural patterns of the posterior left side of the skull are very similar to *Compsocerops cosgriffi*.

**The Palate** (Fig. 4)

*Kumar & Sharma (2019)*, (Fig. 7) described the palate (RH01/Pal/CHQ/Tiki/15) as a metoposaurid palate (Fig. 7). However, no detailed osteological description or identifying characters were described by the authors as to why the specimen was identified to be a metoposaurid. The authors only described the palate to be the 'dorsal' part of a metoposaurid as it has conical teeth present on the anterior part. However, this description is vague and of no taxonomical significance whatsoever as all temnospondyls have conical teeth and both chigutisaurids and metoposaurids have teeth and tusks in the anterior part of the skulls. Again, dentition restricted to the anterior margin of the skull is a characteristic of all temnospondyls with parabolic skulls. Additionally, *Kumar & Sharma (2019)* grouped the palate collected from the village of Tenduadh with the specimens of metoposaurid clavicle collected from the village of Jora. This grouping is not viable as the two villages are approximately 12 kilometres apart from each other and there is a probability that these two villages might be parts of the Upper and Lower parts of the Tiki Formation and may even be of different ages as discussed later. The specimen, as said in *Kumar & Sharma (2019)*, was too friable and could not be excavated by them. Thus, there is no option to study the specimen first-hand. Henceforth, the image of *Kumar & Sharma (2019)*, has been replicated into a high-resolution photograph with the required permission from the editor of the Journal of the Palaeontological Society of India to study the detail of the described specimen (RH01/Pal/CHQ/Tiki/15) (Fig. 7).

The studied specimen is the palate of a temnospondyl and not the dorsal view of the skull as erroneously stated by *Kumar & Sharma (2019)*. The palate showed in the picture (RH01/Pal/CHQ/Tiki/15) (Fig. 7) has a distinct vaulted pterygoid, parabolic skull outline, and comparatively narrow cultriform process than metoposaurids. The specimen (RH01/Pal/CHQ/Tiki/15), (*Kumar & Sharma, 2019*) (Fig. 7) has a parabolic skull, thickening of the pterygoid, presence of vaulted pterygoid, presence of post-pterygoid process and concave vertical wall of pterygoid that are characteristic of chigutisaurids as written repeatedly above. The specimen is indeed friable with dense

networks of fractures that obscured the clear identification of the sutures. The specimen (RH01/Pal/CHQ/Tiki/15) is partly eroded along the lateral margins as well as anteriorly and posteriorly. The right half of the palate is slightly compressed, deformed and curved (Fig. 7).

The anterior portion of the palate is considerably broken both anteriorly and anterolaterally. Though the sutures cannot be delineated, the presence of vomer is very apparent. The vomer is broken anteriorly and the anterior palatal vacuity is not preserved (RH01/Pal/CHQ/Tiki/15) (Fig. 7). The posterior part of the preserved vomer includes the anterior tongue of the cultriform process of the parasphenoid. The left half of a possible vomerine cavity is preserved. The vomerine cavity is present only in Jurassic chigutisaur *Siderops kehli* (*Warren & Hutchinson, 1983*) and it is absent in *Compsocerops* (*Sengupta, 1995*) or *Pelorocephalus* (*Marsicano, 1999*). The left lateral margin of the right choana is aligned to the left lateral margin of the right interpterygoid vacuity. The ectopterygoids are exposed on the anterolateral margins of the interpterygoid vacuities and are preserved on both sides. The ectopterygoid borders the anterior portion of the subtemporal vacuity inwards. The subtemporal vacuity is wide and broad bordered by the ectopterygoid and the parasphenoid on the inward margin and the quadratojugal, the alar process of the jugal on the outward lateral margin. The dentigerous area is restricted to the anterior region of the palate. The anterior and the anterolateral margins of the palate are broken, and all teeth are not preserved. However, two broken ectopterygoid teeth can be seen preserved at the anterolateral corner of the ectopterygoid in contact with the palatine in the left part of the palate. The palatine teeth row in the left half of the skull is also preserved partially. Like other chigutisaurids, the dentigerous area of the palate is remarkably short. The palatine row of teeth is not continuous up to the middle of the choana. This character has been considered to be a synapomorphy of *Compsocerops cosgriffi* (*Sengupta, 1995*). Conical, inward curved four complete palatine teeth are preserved in the margin of the left palatine bone of the palate. Since, the dentigerous area is restricted to the anterior part of the skull the posterior part is longer in proportion and covered by large and wide subtemporal vacuity (Fig. 4).

In (RH01/Pal/CHQ/Tiki/15) (Fig. 7) (*Kumar & Sharma, 2019*) both the interpterygoid vacuities are well preserved. The interpterygoid vacuities are quadrangular in shape, shorter and wider compared to *Compsocerops cosgriffi* (*Sengupta, 1995*). The borders of the interpterygoid vacuities are approximately parallel-sided. The interpterygoid vacuities are bordered dominantly by the cultriform process along the inward margin as well as the vomer. Anteriorly, it is bordered by the vomer and the palatine. The pterygoid forms the dominant margin of the interpterygoid vacuities laterally with a small area being occupied by the ectopterygoid. Posteriorly, these are formed by the parasphenoid. In ISI A 202/1, the interpterygoid vacuities are not completely preserved. In both, the specimens ISI A 202 and (RH01/Pal/CHQ/Tiki/15) (*Kumar & Sharma, 2019*), the interpterygoid vacuities are shorter and broader than *Compsocerops cosgriffi* where the base levels of the interpterygoid vacuities are lower than that of the subtemporal vacuities. The subtemporal vacuity extends anteriorly to the level higher than the centre of the interpterygoid vacuities.

In (RH01/Pal/CHQ/Tiki/15) (*Kumar & Sharma, 2019*) (Fig. 7) both the pterygoids are preserved. They are deep and vaulted. The vertical lateral wall of the pterygoid projects posteriorly possibly up to the posterior level of the occipital condyles which are broken. The palatal ramus of the pterygoid is visible on both sides in (RH01/Pal/CHQ/Tiki/15). The palatal ramus of the pterygoid is longitudinally concave with a gentle fold which is again a character of some chigutisaurids specially *Compsocerops*. The quadrate ramus of the pterygoid is better preserved on the right side of the palate (RH01/Pal/CHQ/Tiki/15). The quadrate ramus of the pterygoid looks like a wall as they are deeply vaulted. The ascending ramus of the pterygoid is not visible in (RH01/Pal/CHQ/Tiki/15). A broken post pterygoid process that is a projection from the posterior border of the pterygoid corpus is visible on the right side of the palate (RH01/Pal/CHQ/Tiki/15). This area on the left side of the palate of (RH01/Pal/CHQ/Tiki/15) is broken. The postpterygoid process is considered to be an apomorphic character for *Compsocerops cosgriffi* (*Marsicano, 1999*). The suture of the quadrate and pterygoid is present on the outer side of the downturned part of the quadrate ramus of the pterygoid. In ISI A 202, (Fig. 4) only the right pterygoid is ill-preserved but the bone surface is crushed. However, a distinct post pterygoid process characteristic of *Compsocerops* is present. Though the bone is crushed and compressed, the vaulted nature of the pterygoid can be made out because of the concavity of the vertical wall of the pterygoid. In both, the specimens (RH01/Pal/CHQ/Tiki/15) (*Kumar & Sharma, 2019*) and ISI A 202, the palatine ramus of the pterygoid is much broader and wider than that in *Compsocerops cosgriffi*.

Just like other chigutisaurids, the base of the parasphenoid is almost hexagonal with a long extension in the form of the cultriform process placed between two interpterygoid vacuities in (RH01/Pal/CHQ/Tiki/15) (*Kumar & Sharma, 2019*). The parasphenoid has a long suture with the pterygoid laterally and the exoccipitals posteriorly. A distinct raised longitudinal keel is present on the ventral surface of the cultriform process in this specimen. The presence of this keel in the cultriform process has been noted by *Marsicano (1999)* as a distinguishing character present only in *Pelorocephalus mendozensis*. However, first-hand studies reveal that this longitudinal keel of the cultriform process is also present in *Compsocerops cosgriffi* from the Maleri Formation of Pranhita, Godavari Valley Basin. The cultriform process of parasphenoid of this specimen is comparatively narrower than all other specimens of *Compsocerops cosgriffi*. The cultriform process of the *Compsocerops* species from Tiki is wider than *C. cosgriffi*, the cultriform process is also comparatively broader than *Siderops kehli*, more comparable to the width of the cultriform process in the specimen previously denoted as *Kuttycephalus triangularis* (*Sengupta, 1995*). The cultriform process preserved in (RH01/Pal/CHQ/Tiki/15) (*Kumar & Sharma, 2019*) is thin and constricted in the middle part of the interpterygoid vacuities and gets broader as it progresses to the anterior part of the process. This type of cultriform process is unique among the chigutisaurids. In the specimen photographed by *Kumar & Sharma (2019)*, the anterior tongue of the cultriform process is in contact with the vomer and lies posterior to the level of the anterior margin of the interpterygoid vacuities. The cultriform process is not preserved in ISI A 202. The occipital condyles are broken as well. In the earliest known

chigutisaur *Keratobrachyops*, the cultriform process of the parasphenoid is also narrower than ISI A 202.

**The Occiput** (Fig. 5)

The occiput is very ill preserved only in ISI A 202 (Fig. 5). The occiput could not be prepared due to the extremely fragile nature of the skull. Removing the matrix load from the occiput would result in the sagging of the entire specimen. However, from the little that could be studied, it can be said that in occipital view, the quadrate ramus of the pterygoid is deeply downturned. The vagus nerve foramen is preserved on the left exoccipital lateral to the broken occipital condyle. The ascending process of the exoccipital is wide and inclined and meets the descending process of the postparietal. A sub-circular, matrix filled, paraquadrate foramen is present in the quadratojugal. The quadrate is partially preserved in the occipital view. It is bounded by the squamosal, quadratojugal and the downturned pterygoid. The absence of occiput makes the comparison of ISI A 202 difficult with the different species of *Pelorocephalus* as different species of the genus are differentiated, to a great extent, by their occipital characters (*Marsicano, 1999*).

**Clavicle** (Fig. 6)

An almost complete left clavicle (ISI A 202/2) (Fig. 6) was found associated with the skull (ISI A 202/1) during excavation. The clavicle has a flat eye-drop shaped blade and a long straight dorsal process that ascends almost straight, nearly at ninety degrees with the plate. The cross-section of the process at the dorsal end is bean shaped as a feeble furrow runs along the process. This is very similar to the clavicle of *Compsocerops cosgriffi* (*Sengupta, 1995*), *Siderops kehli* (*Warren & Hutchinson, 1983*) and *Koolasuchus cleelandi* (*Warren, Rich & Vickers-Rich, 1997*).

## RESULTS

### Tiki Formation

To date, no temnospondyl fauna has been recorded from the upper part Tiki Formation. Excavation taken up in 2018 by the authors revealed the first chigutisaurid from the Tiki Formation in the Tenduadh locality (Fig. 8) in the upper part of the Tiki Formation. Several vertebrae and postcranial bones of metoposaurid have been excavated from the Jora and Tiki Nala sections which have been assigned to the lower Tiki Formation from our field studies. Based on the changes in the faunal assemblage in the lower and the upper Tiki Formation (Tables 2A and 2B) and considering the lithostratigraphy, a boundary between the basal and upper Tiki Formation has been assigned and the zone demarcating the Carnian to Norian faunal turnover in the Tiki Formation has been approximated and marked in red dotted line in the map (Fig. 1). The lithological logs modified after *Kumar & Sharma (2019)* and *Mukherjee & Ray (2012)* reveal that just like the Maleri Formation, the basal Tiki Formation is dominated by a large band of red mudrock intercalated with peloidal calcirudite-calcarenite (*Sarkar, 1988*) (Fig. 9). The Jora Nala section in the Carnian basal Tiki has been logged in detail in this work (Fig. 9A). This shows the dominance of greenish to reddish siltstones and mudstones in the Jora Nala section with sparse deposition of trough cross-bedded sandstones in between.

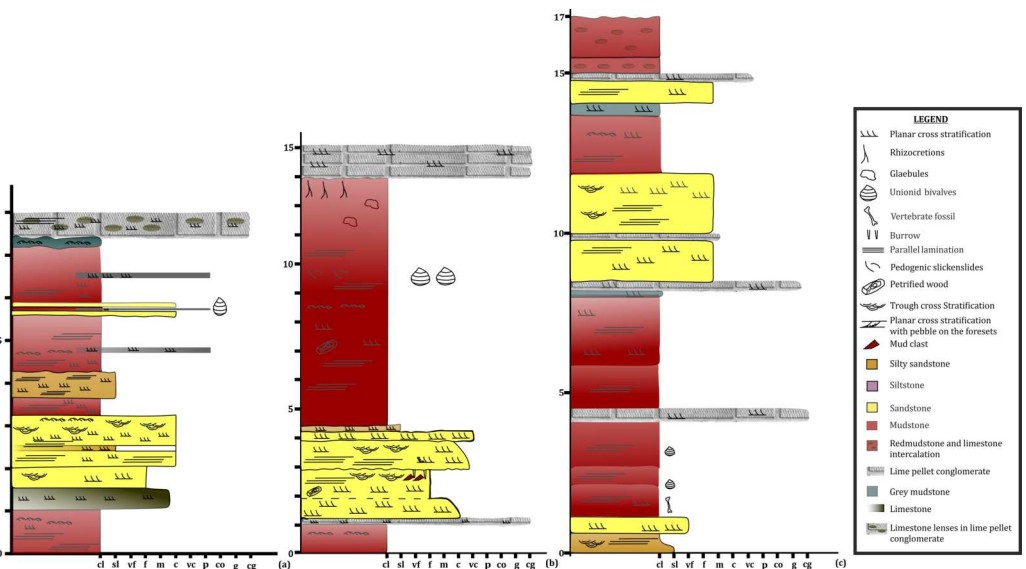

**Figure 8 Lithologs of the Tiki Formation.** (A) Litholog in the Jora Nala Section in the Lower Tiki Formation. (B) Litholog in the Lower part of Tiki Formation modified after *Mukherjee et al. (2012)*. (C) Litholog of the Tiki Formation modified after *Kumar & Sharma (2019)*.

## Maleri formation

In the revised map of the Maleri Formation and (after *Dasgupta, Ghosh & Gierlowski-Kordesch, 2017*; *Kutty & Sengupta, 1989*) a boundary between the Carnian basal Maleri and Middle Carnian/Norian Upper Maleri has been established from both lithological and faunal contents (Fig. 2–faunal boundary indicated by a green broken line). From our field studies and maps it is evident that though sandstone–mudstone alternation is present throughout the Maleri Formation, the Carnian basal Maleri is abundant in red mudrocks and calcirudites (Figs. 2 and 10) and moving towards Upper Maleri there is a sudden increase in the deposition on siliciclastic sediments leading to the more frequent occurrence of sandstone bands alternating with red mudstone (Fig. 10).

## DISCUSSION

The CPE had a significant impact on the terrestrial environment and ecosystem globally. The evidence of CPE has never been worked upon or mentioned in India because of the lack of proper age constraints present in the Late Triassic Maleri and Tiki Formations in India.

### Tiki Formation

No detailed sedimentological or geochemical studies have been carried out in the Late Triassic Tiki Formation in India to analyse the associated changes from Carnian to Norian through the humid phase of the Carnian Pluvial Episode. Though, *Ahmed & Ray (2010)* presented a geochemical analysis of 42 nodular carbonates confirming their pedogenic

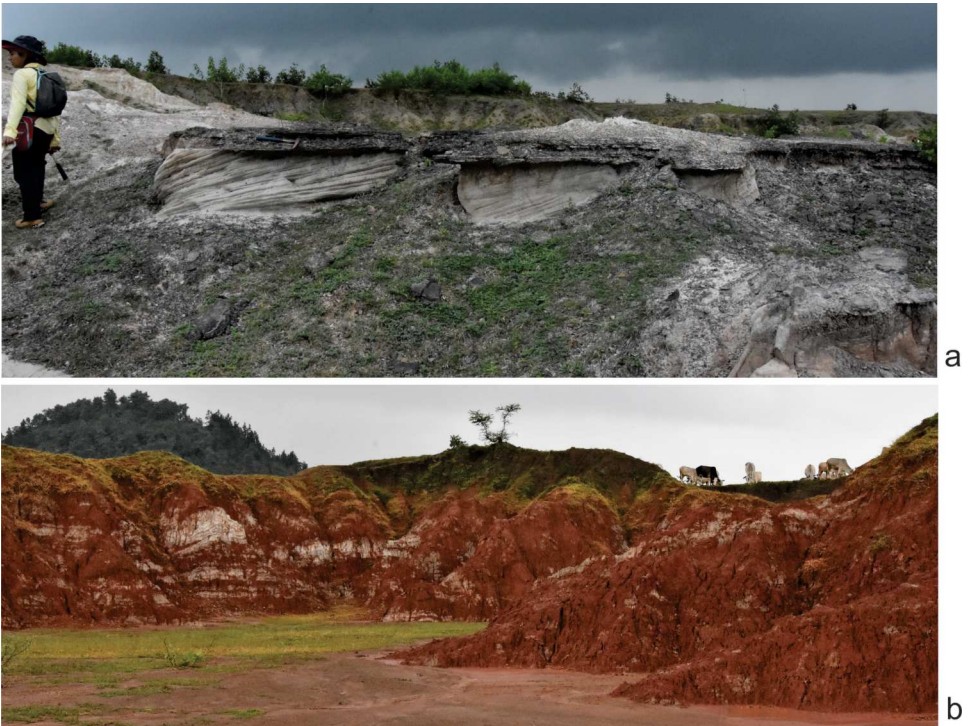

**Figure 9** **Field photograph of the sand-mud alternation in the Tiki Formation near Tenduadh.** (A) Trough cross-bedded sandstone in the Tiki Formation. (B) Extensive mudstone in the Tiki Formation. Photographs taken by the authors during field season 2019.

origin, no details of the localities of collection in terms of lower and upper Tiki have been provided.

The terrestrial influx of sediments is significantly low at the period denoted by the sparse occurrence of sandstones in the basal Tiki Formation (Fig. 9). The presence of Unio beds in between the basal thick layers of mud reflects a stagnant quiet and well-watered environment. This basal mud encompasses areas like the Jora and Tiki River sections. Abundant postcranial fragments of metoposaurids and rhynchosaurs have been collected from these sections. Moving upwards in the direction of the dip of the beds, there is a sudden influx of siliciclastic sediments marked by thick sandstone units with little intermittent mudstone. This could be a demarcation of the rapid influx of siliciclastic sediments that took place during CPE in the Tiki Formation. Only two dominant sand bodies are observed in Tiki before the recurrence of a thick horizon of mud and subsequently sand-mud alternations indicating the onset of seasonality and aridity in the Norian. The Norian of the Tiki Formation is demarcated by red mudstones, whitish sandstones and sparse calcirudites. The Norian Upper Tiki Formation is exposed in sections near Tenduadh as shown in the map (Fig. 1) and an estimated approximate faunal boundary between the Carnian and the Norian in the Tiki Formation is also furnished as in Fig. 1. Tiki has a long history of yielding fossil vertebrates (*Bandyopadhyay & Ray, 2020*; *Chatterjee & Roy-Chowdhury, 1974*) (Table 1). It has a rich Late Triassic faunal association as shown in Tables 2A and 2B

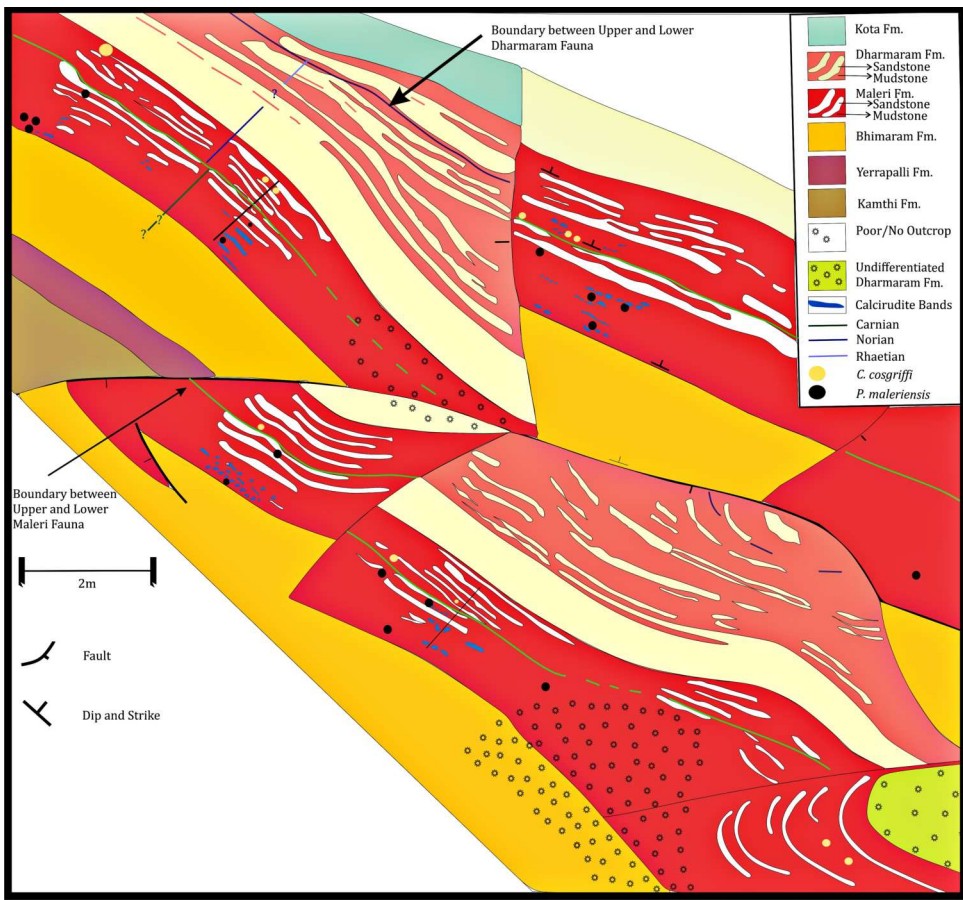

**Figure 10** **Geological map of the Maleri Formation, Pranhita-Godavari Valley Basin, India.** Geological map modified after *Kutty & Sengupta (1989)* and *Dasgupta, Ghosh & Gierlowski-Kordesch (2017)* showing the sand-mud alternations in the Maleri Formation, Pranhita-Godavari Valley Basin, India. The green line represents the faunal boundary that occurred due to the faunal turnover from the Carnian Lower Maleri to the Norian Upper Maleri Formation.

(*Bandyopadhyay & Ray, 2020*; *Chatterjee & Majumdar, 1987*; *Mukherjee & Ray, 2014*). The Tiki faunal assemblage was thought to be coeval to the Lower Maleri faunal assemblage (*Datta, 2005*; *Kutty & Sengupta, 1989*). However, *Datta, Ray & Bandyopadhyay (2019b)* stated that the Tiki fauna ranges from Middle Carnian to Early Norian and is younger than Lower Maleri Fauna. The mid Carnian / Norian Upper Maleri fauna has chigutisaurids. The discovery of a chigutisaurid from the upper part of the Tiki Formation confirms the views of *Datta, Ray & Bandyopadhyay (2019b)* regarding the presence of Middle Carnian/Norian fauna in the Tiki Formation.

## Maleri Formation

The abundance of red mudstone in the basal Maleri Formation (Fig. 11) with a sudden increase in the frequency of sandstone bands in the upper part can be correlated with the advent of the Carnian Pluvial Episode (CPE) in India. The overall palaeoenvironment and sedimentology of the Maleri formation have been worked upon by earlier workers

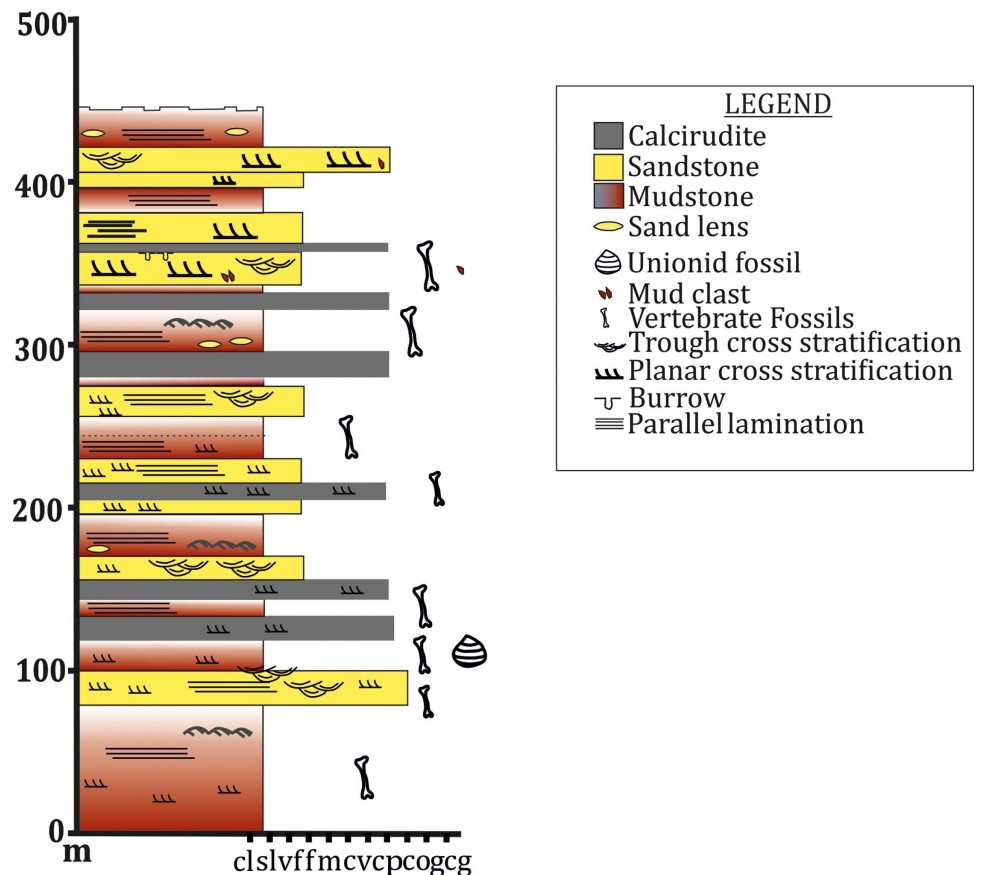

**Figure 11** **Litholog of the Maleri Formation modified from** *Kutty & Sengupta (1989)*.

(*Dasgupta & Ghosh, 2018*; *Sarkar, 1988*). Most of these studies were done on the Maleri Formation as a whole without distinguishing its basal and upper parts. Also, no detailed analyses about the changes in sedimentology or geology or geochemistry have been done to study the changes in the pattern of sedimentation from the Carnian basal Maleri to the Norian Upper Maleri.

Evidence from detailed geological mapping, logs and associated faunal turnover in the Late Triassic Maleri Formation all point toward the distinctive effect of CPE in India. The overall thickness of the Maleri Formation is about 350-600m which is variable in different places. From the map of the Maleri Formation and log of the same near Achlapur and Nalapur (Fig. 10), it is evident that the basal part of the Maleri Formation has a large stretch or band of red mudrock-dominated unit. This unit of red mudrock stretches from about 100m-200m in thickness in basal Maleri and is the thickest band of red mudrock encountered in the entire Maleri Formation (Fig. 12). It encompasses villages namely Aigerrapalli (19°15′22.4″N; 79°27′23.4″E), Achlapur (19°09′34″N; 79°31′51″E) and Nambala (19°13′47″N; 79°26′07″E), areas slightly north of village Gampalpalli (19°10′11″N; 79°30′53″E) and is rich in vertebrate fossils, the most significant among them being rhynchosaurs and metoposaurids. This significantly thick basal Maleri mudstone has
sporadic carbonate grainstones (*sensu Dunham, 1962*; calcarenites–calcirudites of *Sarkar, 1988*) and the presence of palaeosols with no significant siliciclastic deposition of sandstone present within this mud. The upper part of the formation is approximately 200–350 m thick and has three to four multi-storeyed sheet sandstone bodies (each 10–35 m thick) vertically separated by mudrock-dominated intervals (15–55 m thick). The mudrock intervals in the upper part are lithologically similar to those occurring in the lower part of the formation. The mudrock units comprise both stratified and massive mudstones (*Dasgupta, Ghosh & Gierlowski-Kordesch, 2017*) with sporadic carbonate grainstones (calcirudite of *Sarkar, 1988*). Moving upwards from the thick stretch of mudrock, the influx of siliciclastic sedimentation increases as evident from the increase in the deposition of frequent sandstone units. The beginning of these sandstone units is marked by the presence of metoposaurids and rhynchosaurs and unionid fossils in a sandy zone. In the upper part, apart from the chigutisaurids, there are basal sauropods like *Jaklapallisaurus*, and sauropodomorphs like *Nambalia*, probable Guaibasaurids-and-Herrerasaurus-like forms (*Novas et al., 2010*). The authors in the same work, also mentioned that early theropods are known from the Norian-Rhaetian time from North and South America, India, South Africa, and Europe and the demise of members of Lower Maleri fauna like rhynchosaurs together with the global extinction of *Chiniquodon* (cynodont) and Proterocampsidae (archosauriform) mark the Carnian–Norian boundary and also the North Tethyan Pluvial Episode of end Carnian (CPE). *Benton, Bernardi & Kinsella (2018)* have argued that CPE triggered the diversification of early dinosaurs. It has been discussed earlier those indications of CPE are present in Maleri and Tiki Formations of India. The sudden appearance of several basal dinosaurs like *Jaklapallisaurus*, *Nambalia*, probable Guaibasauridae and *Herrerasaurus* (*Novas et al., 2010*) in the Norian Upper Maleri fauna also corresponds to that.

Thus, the palaeoenvironment of the Maleri Formation shifted from a comparatively arid and dry climate in the Julian at the basal substage of Carnian to a high competence fluvial-lacustrine environment with the presence of small, ephemeral and vegetated swamps or ponds along the flow path of the channels at the time of Carnian Pluvial Episode from the end of Julian to Tuvalian and back again to fluvial deposition in the Norian (*Dasgupta, Ghosh & Gierlowski-Kordesch, 2017*). The episodes of increased rainfall during the Carnian Pluvial Episode demarcated by increased frequency of sandstone deposition are intervened by seasonality as evident from the red mudrock alternations between sandstones. Similar climatic shifts are seen from the coeval Santa Maria to Caturrita formations of Brazil (*Dal Corso et al., 2015*). These shifts indicating a major variation of the hydrological regime in terrestrial depositional settings suggest an enhancement of the hydrological cycle during the CPE. Recently, *Lucas (2010)* stated that the demise of metoposaurids in most parts of the world during the Carnian is related to the end of the enhanced hydrological cycle at the dying phase of CPE. The disappearance of key herbivorous groups such as dicynodonts and rhynchosaurs of Carnian and their places taken up by giant sauropodomorphs seems to be linked to CPE which is not documented in India so far.

Also, the *Hyperodepadon* Assemblage Zone (HAZ) is characterized by the presence of rhynchosaur *Hyperodepadon* and is present in the lower part of the Ischigualasto Formation of Argentina, the Lossiemouth Sandstone Formation of Scotland, and the Lower Maleri
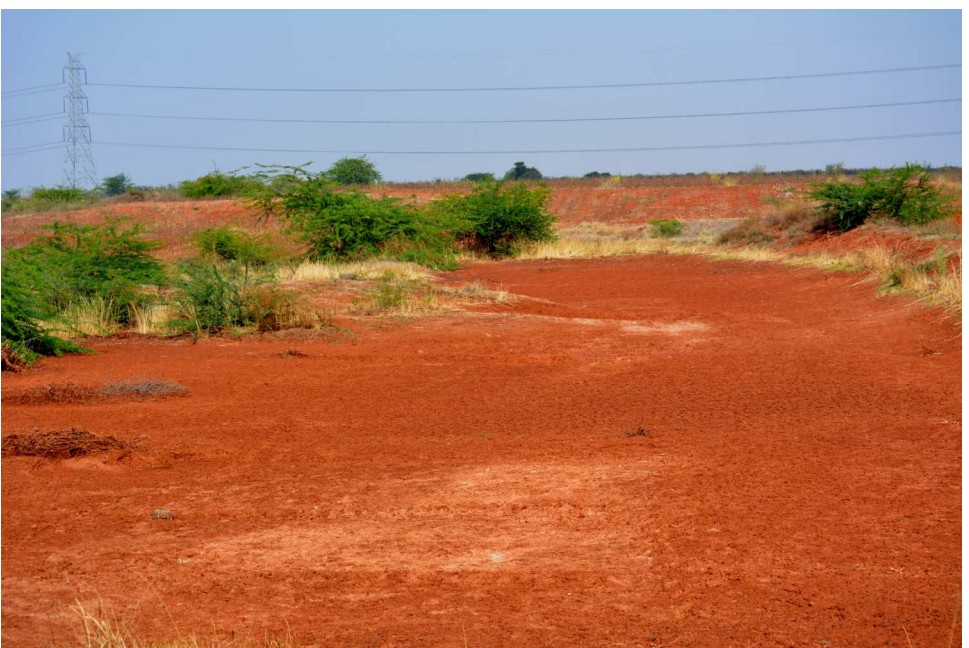

**Figure 12  Field photograph showing the abundance of red mudstone in the basal part of the Maleri Formation.** Image taken by the authors during field season 2015 at Pranhita-Godavari Valley.

Formation of India (*Langer et al., 2010*). The HAZ is dated as late Carnian to early Norian, approx. 228–224 Ma by some authors (*Benton, Bernardi & Kinsella, 2018*; *Brusatte et al., 2010*; *Ezcurra et al., 2017*). Most metoposaurids in the Gondwana deposits are considered to be Carnian in age (*Chakravorti & Sengupta, 2019*; *Gee & Jasinski, 2021b*; *Sengupta, 2002*). The demise of the metoposaurids *Panthasaurus maleriensis* (*Chakravorti & Sengupta, 2019*) in India along with the demise of *Hyperodepadon* (*Chatterjee, 1974*; *Mukherjee et al., 2012*) in both Late Triassic Maleri and Tiki Formation also points to the presence and effect of CPE in India. The demise of the metoposaurids left a vacant niche to be occupied by the chigutisaurids in the Norian suggesting short-lived aridity in post Carnian stage.

The Carnian of Argentina has its age radiometrically constrained between 231.4 ± 0.3 and 225.9 ± 0.9 Ma (*Martínez et al., 2016*) and similar reports are present from Santa Maria and Caturrita Formations of Brazil (233.2 ± 0.7 and 225.4 ± 0.4 Ma) (*Langer, Ramezani & Da Rosa, 2018*). The onset of CPE is well constrained in stratigraphic sections like the Southern Alps of Italy, Northern Calcareous Alps of Austria, Transdanubian Range of Hungary, and the Nanpanjiang Basin of the South China block and is placed at the substages Julian 1–Julian 2 boundary of the Carnian (*Gallet et al., 1994*). Due to the lack of any radiometric dating, Late Triassic Maleri and Tiki Formations are poorly constrained and pose difficulty in global correlation, their correlation is based only on available fauna. The CPE has always been dated as mid-Carnian (*Ruffell, Simms & Wignall, 2016*) but this is not a unanimous viewpoint. The Italian Dolomites occur between the Aonoides/Austriacum interval (about Julian) and the base of the Subbullatus Zone (Tuvalian), dated at 234–232 Ma (*Dal Corso et al., 2015*; *Roghi et al., 2010*). Further constraint has been documented

in borehole successions in the southwest UK, which indicates a maximum duration of 1.09 MYA (*Miller et al., 2017*). The precise radiometric dating to constrain the Maleri and Tiki Formations and to denote the beginning of CPE in India will shed further light on the pattern of faunal diversification post CPE event in the subcontinent and help in the global stratigraphic correlation. A continental carbon isotope record in southwest England shows multiple carbon cycle perturbations during CPE (*Miller et al., 2017*). The CPE is not only the time interval of increased humidity but also a major carbon perturbation. Unfortunately, no carbon isotope data is noted from the Maleri and the Tiki Formations of India.

## CONCLUSION

1. In the current work a new species of chigutisarid amphibian, *Compsocerops tikiensis* sp. nov. from the Late Triassic Tiki Formation of the Rewa Gondwana Basin has been described in detail. The presence of chigutisaurid *Compsocerops tikiensis* sp. nov. in the upper part of the Tiki Formation is the first evidence of the Norian chigutisaurid amphibian from the said Formation and is important for correlation of the Late Triassic basins worldwide.

2. The presence of *Compsocerops* (*Sengupta, 1995*) in Tiki, for the first time, confirms the presence of the Upper Maleri faunal element in Tiki. *Lucas (2020)* thought that the demise of metoposaurids in most parts of the world was at the end of Carnian and that tallies with the last appearance datum of the metoposaurids of Maleri. The chigutisaurids, both in Maleri and Tiki have their first appearance datum at the onset of the Middle Carnian or Early Norian.

3. Along with the extinction of the rhyncosaurs and *Parasuchus* (primitive phytosaur) (*Chatterjee, 1978*), chiniquodontids (cynodonts), the Carnian–Norian Extinction Event (CNEE) also caused the extinction of the metoposaurids in India. Chigutisaurids appeared in Middle Carnian/Norian and India is the only place which accommodates definite metoposaurids and chigutisaurids within the same formations (the Late Triassic Maleri and Tiki Formations) the former being replaced by the latter. Incidentally, among the phytosaurs, the *Parasuchus* of Lower Maleri fauna is replaced by the *Leptosuchus* like forms of Upper Maleri and *Volcanosuchus statisticae* (*Datta, Ray & Bandyopadhyay, 2021b*) in the upper part of the Tiki Formation.

4. The post-CNEE empty niche left by the metoposaurids in the Late Triassic Gondwana deposits of India (controversially Brazil as well, (see *Dias-da Silva, Cabreira & Da Silva, 2011*)  was occupied by the chigutisaurids in the Norian. The availability of phytosaur teeth along with *C. tikiensis* sp. nov. only indicated their co-existence in the same aquatic niche but does not necessarily point towards any prey-predatory relationship between the phytosaurids and the chigutisaurids. However, detailed studies on histology and growth pattern of the chigutisaurids might shed light on the gigantism of these amphibious animals in the post-CNEE and recovery of the temnospondyls.

5. The presence of both metoposaurids and chigutisaurids and the faunal turnover from the Carnian to the Norian along with the extinction of the rhynchosaurs

(*Hyperodapedon*) (*Mukherjee et al., 2012*) and *Parasuchus* (*Chatterjee, 1978*) in the Carnian of both the Late Triassic Maleri and Tiki Formation and the presence of prosauropods in the Upper Maleri Formation and undescribed dinosauriformes including theropod-like forms (*Bandyopadhyay & Ray, 2020*) sheds light and documents for the first time the existence and effect of the Carnian Pluvial Episode in India.

6. The finding of *Compsocerops tikiensis* sp. nov. from the Tiki Formation and assessing its importance in global palaeoclimatic and palaeoclimatic correlation paves the way for future scope of works in finding the effect of CPE in India and comparing its faunal diversification at a global scale. Also, finding any new and better-preserved specimen of *Compsocerops tikiensis* sp. nov. from the Tiki Formation would help to establish the phylogenetic relationship of the other chigutisaurids around the world with that of the Indian counterparts. As the specimens are deformed broken and have poor preservation potential, phylogenetic analysis is beyond the scope of this article as it will add more missing and misinterpreted data.

## ACKNOWLEDGEMENTS

The author SC would like to thank Ms. Aindrila Roy, the Project Linked Person (PLP) at the Geological Studies Unit, Indian Statistical Institute, Kolkata for her help in editing and formatting the manuscript. The authors' SC and DPS would like to acknowledge Mr Lakshman Mahankur for helping in the preparation of the poorly preserved material. We acknowledge the editor (Surendra Kumar) of the Journal of Palaeontology of India for providing us with permission to reproduce the published picture published on page 159, (Plate IV. number 1) of the journal volume 64(2), December 31, 2019. We are also very grateful to Dr Claudia Marsicano, Dr Valentin Buffa and the anonymous reviewer who through their insightful comment and meticulous suggestions has enriched this work to a great extent.

### Funding

This work has been supported by the Indian Statistical Institute, Kolkata, an Institute of National Importance under the Ministry of Statistics and Programme Implementation (MoSPI), Government of India. The funders had no role in study design, data collection and analysis, decision to publish, or preparation of the manuscript.

### Grant Disclosures

The following grant information was disclosed by the authors:
Indian Statistical Institute, Kolkata.
Institute of National Importance under the Ministry of Statistics and Programme Implementation (MoSPI), Government of India.

## Competing Interests

The authors declare there are no competing interests.

## Author Contributions

- Sanjukta Chakravorti conceived and designed the experiments, performed the experiments, analyzed the data, prepared figures and/or tables, authored or reviewed drafts of the article, and approved the final draft.
- Dhurjati Prasad Sengupta conceived and designed the experiments, performed the experiments, authored or reviewed drafts of the article, and approved the final draft.

## Data Availability

The raw data for this manuscript are available in the figures and tables.

## New Species Registration

The following information was supplied regarding the registration of a newly described species:

Publication LSID: urn:lsid:zoobank.org:pub:1B45D1E1-9FFE-421D-8060-4174334A7EF4

Compsocerops tikiensis LSID: urn:lsid:zoobank.org:act:B2BC0270-EB12-4E9A-99B1-8D7248EC1E88

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
