# Peer review of "The first record of chigutisaurid amphibian from the Late Triassic Tiki Formation and the probable Carnian pluvial episode in central India"

_PeerJ, doi:10.7717/peerj.14865_

## Round 0.1 · original submission · Major Revisions

Dear Dr Chakravorti,

I am writing you in relation to your manuscript (#67301) entitled “The First Record of Chigutisaurid Amphibian from the Late Triassic Tiki Formation and the probable Carnian Pluvial Episode in Central India” co-authored with D. Sengupta.

The manuscript has been reviewed by two reviewers and myself and after a careful evaluation, I consider your manuscript needs Major Revision before it can be considered for publication in our journal.
The two reviewers commented extensively about the manuscript, and in any case, both concur that the manuscript is interesting and deserves to be published.

Below, I point out some concerns shared by the reviewers and myself, that I would like you to pay particular attention:

Following the suggestions made by both reviewers, I recommend rearranging the manuscript sections:

- Introduction: include information about the group (Chigutisaurudae) in a wider context as distribution (temporal and spatial), comments of their phylogeny, and then why they were/are important in the context of Indian Triassic biostratigraphy. Also, you should include here the information in the “Previos Work…” section of the manuscript.
Also, I suggest including a subsection, or you can make a new one (Geological Setting), with all the geological information on the Tiki and Maleri formations (stratigraphy, spatial distribution of the outcrops, paleontological content, etc. etc.) to avoid repetition along the Ms of the same data.

- Materials and Methods

- Systematic Paleontology: include here also the additional chigutisaurid specimen.

- Discussion: you should include also a paragraph describing/explaining the CPE in a wider context before you discuss its impact in India.

Be aware you should change the figure´s order in order to match the new suggested structure of the manuscript.

Also, I would also recommend that Table 1 be separated into two tables, one for the fish content and another for the tetrapod content. Table 1 in its present format is rather confusing.

Finally, the figures need a thorough overhaul, please check them carefully as they have many errors: Figures 1, 2 and 3 must have the same layout, including the size and shape of the lettering, and quality of the line-drawings. Figure 5 is out of focus; please provide a better photograph and also include the orientation of the bone in its epigraph. Figure 8 needs to be redrawn. All the references to the column sections should be merged in only one to avoid repetition and should be larger so they would be easier to read. Also, the individual letters identifying the sections are all different. Figure 11, please locate the references on the right side of the section to avoid blank spaces; in the references, what are you indicating with the “Fossil” icon? Because I presumed the unionid bivalves are also fossils, so please clarify.

I am requesting that you revise your manuscript according to the detailed reviews provided, taking particular attention to the points mentioned above.

As the revisions required are extensive enough, another round of review may be necessary when you resubmit your revised manuscript.

Thank you for submitting your work to PeerJ and I look forward to receiving your revision.

Sincerely,

Claudia Marsicano

·

Basic reporting

This is a nice descriptive work of very interesting new finds, with a well-documented thorough review of the stratigraphy of the Tiki Formation. However, the identification of the specimens ought to be more argued, and I have minor issues with the organization of the description and stratigraphy sections.

Experimental design

No comment.

Validity of the findings

No comment.

Additional comments

Description and identification
I have several issues with the Materials, Systematic Paleontology and Description sections of the manuscript.

1. I suggest reducing the Materials section to an overview of the preservation of specimens of ISI A 202 on one hand, and specimen RH01/Pal/CHQ/Tiki/15 on the other. Importantly, a case should be made regarding the association of the various sub-specimens of ISI A 202 in a single individual (especially the clavicle). Also, provided the fragments are associated, I suggest that all material confidently interpreted as a single individual should be included in the holotypic material.

2. This manuscript needs a detailed section on the attribution of both specimens to Compsocerops tikiensis. At present, while the authors provide a good argument against the metoposaurid attribution of one specimen, the attribution to trematosaurids, brachyopoids, chigutisaurids, and finally Compsocerops is lacking, with various elements of this identification scattered throughout the manuscript. A dedicated section is much needed in my opinion, either before or after the Description. This section should not be a ‘diagnosis of the genus’ sub-section in the Systematic Paleontology, except if the authors wish to revise the diagnosis of Compsocerops as a whole.
This is particularly important as this new taxon apparently shares unique characters with other chigutisaurids that need to be specifically addressed in the manuscript (e.g. the vomerine vacuity described lines 250-252 is only known in Siderops).

3. I find the description to be somewhat lacking and heterogenous. Some parts, such as the sulci, horns and palate are described in some length, yet the rest of the description mostly consists on comments on the preservation or the identification of RH01/Pal/CHQ/Tiki/15. At least the relative position, size and shape of each bone/bone portion should be described. In particular, some characters in the ‘diagnoses’ are absent or incoherent with the description (e.g. postparietal horns described as ‘putative’ line 157, but with ‘clear evidence’ line 206).

Stratigraphy and CPE
1. While the authors put forward convincing arguments for the identification of the CPE and Carnian-Norian boundary in the Tiki Fm., I found the stratigraphic section of the manusript confusing at times, with discussions coming much later than affirmations (e.g. lines 416-149). This may give the reader the impression of a circular reasoning. I would suggest moving the description of the Maleri Fm. first to have a much clearer comparison with the Tiki later on, but leave this decision at the discretion of the authors. Regarding the Tiki Fm., I feel the identification of the CPE, Carnian and Norian should come after the description and faunal correlation.

2. I understand that the demarcation between the upper and lower beds of the Tiki Fm. is new and might need further work before being formally described as Lower Tiki Fm. and Upper Tiki Fm. However, I suggest choosing a specific terminology and including it in Figure 8 for clarity.

3. Several names are employed for the CPE in the manuscript, with “Carnian Pluvial Event” being the most recurrent. However, as stressed by Dal Corso et al. (2018), this terminology is misleading, and I highly suggest following their terminology of “Carnial Pluvial Episode”.
Dal Corso, J., M. J. Benton, M. Bernardi, M. Franz, S. Hohn, E. Kustatscher, A. Merico, G. Roghi, J. G. Ogg, N. Preto, A. R. Schmidt, L. J. Seyfullah, M. J. Simms, Z. Shi, and Y. Zhang. 2018. First workshop on the Carnial Pluvial Episode (Late Triassic): a report. Albertiana 44:49–57.

References
In the text, the references are ordered alphabetically and not chronologically as usually done and as specified in the PeerJ guidelines: “Multiple references to the same item should be separated with a semicolon (;) and ordered chronologically”. Please correct the main text accordingly.
There are several incomplete references in the reference list, such as Bandyopadhay and Ray (2020), Cotter (1917), Dunham (1962), Kutty et al. (1987), etc. please check all reference.

Some minor comments:
- lines 5-8: affiliations 1 and 2 are the same.
- line 21: remove “too”.
- lines 22-23: some “the” are missing before most geological periods (e.g. “THE Late Triassic”).
- lines 24-27: I don’t understand how the CPE (generally dated ‘mid-Carnian’) can be correlated to the advent of chigutisaurids in India… whose appearance in India mark the Carnian-Norian boundary. Please rephrase as this seems incoherent.
- line 34: remove “the” between “Both” and “Formations”; “metoposaurid” should be singular.
- lines 35-37: put “In the Maleri Formation” at the beginning of the sentence, otherwise this is confusing.
- lines 37-39: several references of the reference list should be cited here.
- line 41: add “as the Lower Maleri Formation” after “as well”.
- lines 44-45: what is the point of this sentence on taphonomy? Please rephrase or remove.
- lines 47-50: both sentences should be fused for clarity.
- line 48: add “The” before “Weathered”.
- line 51: remove reference to Fig. 1a here, as it is confusing.
- line 55: add “the” before “Appearance”; previous studies should be references in this sentence.
- line 61: what does “maximum” mean here?
- line 70: add “the” between “Only” and “left”.
- line 72: change “has” for “have”.
- line 73: add “the” before “Palate”.
- line 98: add “The” at beginning of sentence.
- line 117: remove “in”.
- lines 120-123: this sentence could be removed to avoid repetition.
- line 124: there are too many brackets.
- line 130: change “reptilian” to “reptiles”
- line 133: remove “list of the”.
- lines 137-140: these taxonomic authorities should not have brackets.
- line 195: add “the between “However” and “anterior”.
- line 198: add “. The” between “part” and “parietal”.
- line 208: “These hors are not preserved…” do you mean they are broken or they are absent in those taxa?
- lines 209-210: there is one “(“ too many.
- line 268: this figure refers to ISI A 202 although the description is almost centered on RH01/Pal/CHQ/Tiki/15. Please clarify.
- line 287: “especially” instead of “specially”.
- line 316: can you be sure the unique shape of the cultriform process is not due to breakage?
- line 354: add “The” before “CPE”.
- line 368: add “The” before “Tethys”.
- line 371: add “the” before “CPE”.
- line 378: add “the” before “CPE”.
- lines 381-383: I don’t follow this part.
- line 393: remove “the”.
- line 397: “Formations” plural.
- lines 411-413: this is contradictory, please rephrase.
- line 427: “Unionid” instead of “Unio”?
- lines 448, 450: Datta et al. (2021) instead of Datta et al. (2019).
- line 451: Add “the” before “Presence”.
- line 471-474: this needs references.
- line 521: Lucas (2020) instead of Lucas (2010)?


Table and Figures
- Table 1: “Undescribed” should not be in italics, please correct all occurrences.
- Figure captions: please increase the size of the captions on most figures, especially on the specimens, as they are very small and hard to read.
- Figs. 1&2: “stf” should refer to “subtemporal” fenestra rather than foramen.
- Fig. 1 caption: While the photograph is recreated from Kumar and Sharma (2019), I understand the nice line drawing is new. I suggest rewriting the caption entirely in the format of Fig. 2 and referencing Kumar and Sharma (2019) only in the Fig. 1A sub-caption. Also, is it possible to mention the rough size of the geological hammer on fig. 1A to have an idea of the scale? I understand Kumar and Sharma (2019) gave no indication.
- Fig. 2: I suggest adding captions for the sulci and horns to help understanding the description.
- Figs 2&3: I understand both pertain to the same specimen. However, the scales seem to have doubled from 5cm to 10cm, please check that this is correct.
- Figs. 4&5: several captions could be added to help understanding the description.
- Figs. 8&11: I suggest adding the name of the formation(s) on the logs.

Reviewer 2 ·

Basic reporting

Dear Editor of PeerJ,
Dear Authors,
I have reviewed the manuscript entitled: “The First Record of Chigutisaurid Amphibian from the Late Triassic Tiki Formation and the probable Carnian Pluvial Episode in Central India” by Sanjukta Chakravorti and Dhurjati Prasad Sengupta.

It is a very interesting study presenting a new species of chigutisaurids Compsocerops tikiensis from the Late Triassic of Tiki Formation. It is a very important finding, especially for the knowledge about chigutisaurids. That group of temnospondyls is still one of the less known and each new record helps to understand the evolution of the group. The paper focuses on three main topics:
1. The description of the new material and taxonomical diagnosis
2. New diagnosis of the skull earlier published as a metoposaurus by Kumar & Sharma (2019)
3. The analyze of the Carnian Pluvial Episode in India
Although the issue is really interesting and deserves to be published in a high ranked journal as PeerJ, I found some problems that must be assessed before final acceptation of the paper.
The paper is built from five main chapters: introduction, material and methods, systematic paleontology and discussion. The systematic part with the presentation of the new material is well constructed. All possible to see, in the given state of preservation, anatomical characters are correctly described and marked on the figures. The erection of the new species seems to be justified. Contrary to the systematical part, the introduction and discussion have to be completely reworked. In my opinion it could be very helpful if you will add on the beginning of the introduction some general information about temnospondyls, then present the phylogenetical position of chigutiaurids among all temnospondyls. A short introduction of the knowledge up to date about existing chigutisaurids taxa, their paleogeography and time range will help to understand the importance of your finding. It will follow you to the presentation of the all temnospondyls of India from the critical time spot and the coexistence with chigutisaurids. Then you can add short chapter about CPE, what does it mean, the current stage of knowledge about the event in the world. These facts are important, first to put some structure into your text, and second will help researchers not directly familiar with the chigutisaurids and geology of India to follow your article. The chapter Previous works and Literature should be incorporate to the introduction and not after the material and methods.
Also, the lines 79-86 – you present here the information why Kumar & Sharma (2019) specimen is a chigutisaurud and not a metoposaurus. In my opinion it should be a part of your paleontological taxonomy chapter – you use there that specimen as a referred material. And maybe in that point you should discuss short and provide a new taxonomical diagnosis.
The same comments I have about discussion, it is focused very much on India. I think it is necessary to present the new finding and your results into wider content. The characteristic of the geological situation of two localities is very detailed and of course important to present the CPE but partially the same information you have in previous chapter and you have to clear state the importance of your new findings as a proof of CPE in Central India.
Authors cited all important papers which are published about this topic. However new references will be useful, especially to present the worldwide distribution of chigutisauride and their taxonomical position. Some small errors occur in the formation of the reference list, but they are detail which can be easily corrected. Also, authors sometimes provide sentences with some general statements: ie. lines: 38-40 - Though considerable amount of work has been done on the microvertebrates, rhyncosaurs and phytosaurs of the Tiki Formation, no recent comprehensive works have been done in the last decade on its temnospondyl faunal contents – for such information you need to add references (the considerable amount of work about microvertebrates, rhyncosaurus and phytosaurs).
Figures and table are correct, however small technical issues should be corrected. The format of the scale bars and the size of fonts is for each figure slightly different, the same with the labelling. I think if you have a scale bars with the numeric value on the figures, you do not need to add in the captions again the same information.
Authors used professional English language throughout.

Experimental design

It is an original primary research within scope of the journal. Goals in the chaotic introduction are difficult to find. Of course, the presentation of the new material is a clear goal, but well-established scientific questions will help you to provide a logical discussion. Methods are described with sufficient detail.

Validity of the findings

For sure it is a very important paper and should be published after crucial change of the structure.

Additional comments

The paper is very important for improving our knowledge about temnospondyls. Even, without the CPE background, the establishing of the new taxon makes the paper worth to publish. Authors present a lot of information; however, a little chaotic structure of the manuscript make difficult to follow the main thinking line. I would say, that all facts are inside, maybe except very few general introduction information, but need to be ordered.

---

## Round 0.2 · Major Revisions

Dear Dr Chakravorti,

I am writing you in relation to the R1 of your manuscript (#67301) entitled “The First Record of Chigutisaurid Amphibian from the Late Triassic Tiki Formation and the probable Carnian Pluvial Episode in Central India” co-authored with D. Sengupta.

The R1 has been reviewed by a reviewer and myself and after a careful evaluation, I consider your manuscript needs Major Revision before it can be considered for publication in our journal.

Some changes, particularly related to the format and structure of your manuscript and the figures, still need particular attention. This new version continues raising some concerns related to the Discussion. There is not a “Discussion” section, instead it is atomized in 4 separate first order titles (“The Carnian pluvial episode-A global climatic consequence”, “Significance of temnospondyl amphibians in the Carnian Pluvial Episode”, “Significance of Compsocerops tikiensis in demarcating the Carnian Pluvial Episode (CPE) in India”, and “The Effect of CPE on the terrestrial ecosystem of Maleri and Tiki Formations”) plague of redundant information some already previously mentioned in the text and/or repeated among these 4 sections. Please re-write the whole Discussion, merging the 4 sections, and sharpening and tightening the text. This is an important part of your paper and needs to be better constructed.

Also, Table 2a is rather confusing with “OSTEICHTHYES” at the same level as “Genus and Species”? Please revise its structure carefully.

Still the figures need a thorough overhaul as they are plague of errors:
Figures 1, 2 and 3 still do not have the same layout, including the size and shape of the lettering, and quality of the line-drawings, as was previously pointed out.

Figure 5 is still out of focus; please provide a better photograph and you did not include the orientation of the bone in its epigraph.

Figure 8 still needs to be redrawn. Apparently, when you modified it the lettering and references of the columns (thickness in meters, rock types at the base, and references) resulted in different sizes among the columns, making some of them unintelligible. The references are also too small and there are icons without explanation. Again, in the references, what do you consider a “Fossil”? seems to me that only the vertebrates, but I presumed the unionid bivalves are also fossils, so please clarify?

Figure 10. All the names and letters (references) on the map are of different sizes. They must be homogenously formatted.

Figure 11. The references (icons) should be all the same size along your log as well as the lines between the beds, and the icons must be consistent between the column and the references. Please fix these inconsistences.

I am requesting that you revise your manuscript according to the detailed review provided taking particular attention to the points mentioned above. Also, please specify in detail in your rebuttal letter all that changes here suggested that you are not considering and why.

Thank you for submitting your work to PeerJ and I look forward to receiving your revision.

Sincerely,

Claudia Marsicano

Reviewer 2 ·

Basic reporting

Definitely the part which have to be reworked is discussion. General problem for the discussion is, that it is a mix of new results (or I suppose that based on i.e. line 449: The Jora Nala section in the Carnian basal Tiki has been logged in detail in this work… if you did that in detail and you are using the results of this log, then it is a new result) and external references and it is not always clear which statement is your and which is a citation. Try to separate new results (as ie the description of the geology) from the interpretation. Moreover, it looks a little like two papers, one describing the next taxon with the anatomical description and taxonomical determination, and the second paper is about the geology. Maybe it is not a bad idea to separate both topics in two independent papers – a determination of a new species and in the second you can focus on the geological aspects. The paper now is very long, and if you will add the proper information to material and methods and results (about geology) it will be even longer. But of course, it is only a suggestion – you can leave both topics in one paper, but you have to better link both aspects.

Experimental design

It is an original primary research within scope of the journal. Introduction is now much better, but the first two chapters from discussion should be moved to introduction as they give a general overview of the CPE problem. Few sentences are still not clear – see marked pdf. In Material and Methods add information about the geological part of the paper (the list of new logs) and in results add the description of the geological profiles (they occur for the first time now in the discussion). The discussion should focus only on the comparison between both formations (based on your own results and papers). Also, you have to incorporate the importance of the new skull of chigutisaurid for the comparison between both localities. Moreover, the lines from 435 to 455 should be rewritten – now it looks like the brain-storming notes – try to describe the map, provide why you did that, what new you were able to show, etc?

Validity of the findings

no comment

Additional comments

For the details see attached pdf with comments.

Annotated reviews are not available for download in order to protect the identity of reviewers who chose to remain anonymous.

---

## Round 0.3 · Minor Revisions

Dear Dr Chakravorti,

I am writing you in relation to the R2 of your manuscript (#67301) entitled “The First Record of Chigutisaurid Amphibian from the Late Triassic Tiki Formation and the probable Carnian Pluvial Episode in Central India” co-authored with D. Sengupta.

The R2 has been reviewed by a reviewer and myself and after a careful evaluation, I consider your manuscript has improved substantially and only needs a Minor Revision.

As pointed out by the reviewer, there is a mismatch between the reference list and the text citations, so please make a careful revision of the reference list.
Also related, many of your text citations are incorrectly formatted: both the authors and date are in between brackets when, according to the way the sentence is constructed, only the date should be in between brackets. Please read carefully the text and fix these citations.

Figure 10. still has all the names and letters on the map of different sizes; they should be homogeneously formatted.

In Figure 11. the “Legend” includes “Calcidurite” and “Calcirudite”, both in grey shades. I certainly known calcirudite (light grey in the legend) as a type of limestone, that you also mentioned in the text. But according to your figure it is not present in the section? So, all the dark grey beds are “Calcidurite”? Which is the difference between them?

I would appreciate if you could send to me the new revised version as soon as possible so I can continue with its editorial process.

Sincerely,

Claudia Marsicano

Reviewer 2 ·

Basic reporting

Revision of submitted article (#67301)
The First Record of Chigutisaurid Amphibian from the
Late Triassic Tiki Formation and the probable Carnian
Pluvial Episode in Central India
Dear Editor of PeerJ,
Dear Authors,
I have done a third round of the revision of the manuscript entitled: “The First Record of Chigutisaurid Amphibian from the Late Triassic Tiki Formation and the probable Carnian Pluvial Episode in Central India” by Sanjukta Chakravorti and Dhurjati Prasad Sengupta.
I am happy to see that the current version is good, however still some issues have to be solved, before the publication.

1. BASIC REPORTING
Now the paper is well structured. Only the formatting has to be corrected, as some technical issues still occur.
1. The reference list is not complete:

Konietzko-Meier et al. 2019, Konietzko-Meier et al. 2018, Schoch and Milner 2014 – are missing in the reference list – please make sure that all articles mentioned in the text are in the reference list.

2. Citation format – in many places in the introduction you put both, the names and date in the bracket – even if only the date should be in bracket. I.e.
Lines 61, 62: (Chakravorti & Sengupta 2019) in their taxonomic revision of the Indian metoposaurids, included the metoposaurids - here only 2019 in bracket
Line 66 Formation have been discussed by Rakshit et al. 2020 – as well as here
Lines 117-123. According to (Fortuny et al. 2019) the gigantism of the metoposaurids might have been linked to the Carnian Pluvial Episode. (Buffa et al. 2019) also stated that the diversification of the metoposaurids might have been linked to the CPE and the post-CPE aridification led to the extinction of the metoposaurids during the Rhaetian. (Gee & Jasinski 2021a) have also commented on the fact that the physiological variation of the metoposauridae and their palaeoclimatic range also corroborates a palaeo-environmental barrier. Finally, (Lucas 2020) concluded that climate change that occurred during CPE played an important part in the metoposaurid evolution. – the same problem
Generally, in the introduction there are many problems with the format. Please make sure that it is corrected.

3. Lines 129-130: ... possible reason for faunal turnover from Carnian to Norian (Sengupta 1995) concerning the amphibious temnospondyls. – the name on the end of the sentence

4. Add reference:

Lines 177-180: The chigutisaurids in Maleri appear just above this sandy zone (Sengupta 1995) and no rhynchosaurs or metoposaurids are known from that level (or above that) – REFERENCES – OR SENGUPTA 1995 IS AS WELL VALID FOR THAT- IF YES CITE ON THE END OF SENTENCE. The occurrence of chigutisaurids in Tiki is also restricted within a sandy zone which do not contain metoposaurids or rhynchosaurs (REFERENCE). Unionids are also present there but in lesser abundance than Maleri (REFERENCE). Phytosaur teeth are also present (REFERENCE).

5. Line 196-197: The palate ISI A 202/1, 197 ISI A 202/3-5 (Fig: 4). – something is missing in that sentence

6. Line 506: formatting… dominated unit. This unit of red mudrock stretches from about 100m- 200m in thickness – add space after 100m or delete before 200m

7. Lines 512-513: Maleri mudstone has sporadic carbonate grainstones (sensu Dunham 1962); calcarenites-calcirudites of Sarkar 1988) … something wrong with the brackets, you need to add a second one before sensu, or before 1988
Line 578: Unfortunately, no carbon isotope data is noted from the Maleri and the Tiki Formations of India. – different size and type of font. In pdf I can see similar problems with the size and type of font in many places, however I cannot exclude that it is a result of the transformation of the word document to pdf – but make sure that your MS is correct.
Line 705: DIAS-da-SILVA S, - font format



Best regards and I hope to see the paper soon publish – it is a really nice material and topic.


Thank you!

Experimental design

no comments

Validity of the findings

For sure it is a very important paper and should be published after correcting the technical issues.

Additional comments

no comments

---

## Round 0.4 · Minor Revisions

Dear Dr Chakravorti

After reviewing the new version of your Ms, there are still minor things that need to be corrected. All of them where already mentioned by me and/or the reviewer in previous reviews.

1.- I pointed out: ..." many of your text citations are incorrectly formatted: both the authors and date are in between brackets when, according to the way the sentence is constructed, only the date should be in between brackets. Please read carefully the text and fix these citations..."

E.g. lines 117-126: "....According to (Fortuny et al. 2019) the gigantism of the metoposaurids might have been linked to the Carnian Pluvial Episode. (Buffa et al. 2019) also stated that the diversification of the metoposaurids might have been linked to the CPE and the post-CPE aridification led to the extinction of the metoposaurids during the Rhaetian. (Gee & Jasinski 2021a) have also commented on the fact that....."

These sentences are wrongly constructed as you cannot mentioned directly an author that it is between brackets? you should put only the date between brackets in these cases. This error does nothing to do with Endnote and how PeerJ allows you to submit the references, as you mentioned in your rebuttal letter.

2.-·Figure 10. still has all the names and letters on the map of different sizes; they should be homogeneously formatted.

In your rebuttal letter you said: .."We have tried to homogenize all the letters. However, since the map after Kutty and Sengupta has only a JPEG version, this could not be corrected. This is kind of embedded in the JPEG. The only available copy we have of the map is that of Kutty and Sengupta 1989. Since this map is very important for the later workers we decided to publish it not violating any copyright issues. I would be grateful if you can suggest me a way to homogenize it. Unfortunately, we do not have any software that could homogenize the map from within the JPEG format...."

I consider this is not a good answer....the map can be redrawn (e.g. using Illustrator) and this would tackle the two issues, format and copyright?

3.- I pointed out: "Figure 11. the “Legend” includes “Calcidurite” and “Calcirudite”, both in grey shades. I certainly recognized calcirudite (light grey in the legend) as a type of limestone, that you also mentioned in the text. But according to your figure it is not present in the section? So, all the dark grey beds are “Calcidurite”? Which is the difference between them?

You answered in the rebuttal letter: "We have corrected it"

You just corrected the misspelling so the section references has now two calcirudite, one in light grey and the second in dark grey. How should a reader interpret these two calcirudite ?

In any case, still you have just only one (dark grey) represented in the section but not the other, so what is the point of listing both in the References?

Please, read carefully the comments above and if you have any doubts do not hesitate to contact me before submitting the manuscript.

Sincerely,

Claudia Marsicano

---

## Round 0.5 · accepted · Accept

Dear Dr. Chakravorti,

I am pleased to inform you that your manuscript Ms "The first record of chigutisaurid amphibian from the Late Triassic Tiki Formation and the probable Carnian pluvial episode in central India", co-authored with D. Sengupta, is now accepted for publication in PeerJ.

Thank you again for considering PeerJ and we look forward to your future contributions to the Journal.

sincerely,

Claudia Marsicano